# MATH BLIND: FAILURES IN DIAGRAM UNDERSTANDING UNDERMINE REASONING IN MLLMS

**Yanpeng Sun**[2,*] **Shan Zhang**[1,5,*,†] **Wei Tang**[3] **Aotian Chen**[4] **Piotr Koniusz**[6,5] **Kai Zou**[7] **Yuan Xue**[4‡] **Anton van den Hengel**[1‡]

[1]Adelaide AIML, [2]SUTD IMPL, [3]NJUST IMAG, [4]OSU, [5]Data61♥CSIRO, [6]UNSW, [7]NetMind.ai
shan.zhang@adelaide.edu.au, yanpeng_sun@sutd.edu.sg

## ABSTRACT

Diagrams represent a form of visual language that encodes abstract concepts and relationships through structured symbols and their spatial arrangements. Unlike natural images, they are inherently symbolic, and entirely artificial. They thus pose unique challenges for Multimodal Large Language Models (MLLMs) distinct from natural image processing. Recent studies have shown that MLLMs often exhibit flawed reasoning and hallucinations when handling diagram inputs. We investigate here whether these limitations stem from shortcomings in the models' ability to interpret diagrams themselves. To this end, we develop a diagnostic test suite that isolates perception from reasoning. Our systematic evaluation reveals that MLLMs perform poorly on basic perceptual tasks, *e.g.*, shape classification, object counting, relationship identification, and object grounding, with near-zero accuracy on fine-grained grounding. Further analysis shows that weak diagram perception leads to "blind faith in text", where models rely on textual shortcuts rather than visual understanding (that is, they are *Math Blind*). We hypothesize that enabling models to capture the inherent structural properties of diagrams, represented as graphs of primitives and their interrelationships, is essential for improving diagram understanding. Experiments with 7B and 32B MLLMs validate this assumption, with models trained on such representations achieving a +79% gain on the grounding task. Crucially, these gains transfer to reasoning, achieving 3–4% cross-suite improvements on four public benchmarks even without additional chain-of-thought reasoning data. Our findings demonstrate that low-level perception supports faithful high-level reasoning in mathematical MLLMs. We provide both methodological frameworks and empirical evidence to guide future research in this direction. Our project page is at viocean/MATHEMETRIC.

## 1 INTRODUCTION

Computer vision has traditionally focused on image-based perception tasks such as object detection Meng et al. (2021); Zhu et al. (2020), segmentation Li et al. (2021); Mishra et al. (2019), and spatial reasoning Cheng et al. (2025); Driess et al. (2023); Lin et al. (2014). These capabilities form the foundation for higher-level visual reasoning in recent advances of Multimodal Large Language Models (MLLMs) Achiam et al. (2023a); Liu et al. (2023b); Sun et al. (2024). Despite remarkable successes in general vision tasks, current MLLMs face considerable challenges in interpreting mathematical diagrams.

Although both natural images and symbolic diagrams can be represented as grids of pixels, they constitute very different forms of information. Images represent samples of the intensity of the real world. Diagrams, despite taking many forms, are uniformly generated by humans as abstract visual representations of concepts characterized by precise geometric structures and symbolic notations Lu et al.; Zhang et al. (2024a). Images are natural collections of signals, where diagrams are artificial collections of symbols. Diagrams are critical to human visual communication across educational contexts, scientific discourse, and STEM problem-solving, but the problem of analyzing symbolic visual information is far broader.

---

*Core Contribution    †Project Lead    ‡✉: Yuan.Xue@osumc.edu; anton.vandenhengel@adelaide.edu.au

Recent benchmarks such as MathVista Lu et al. and MathVerse Zhang et al. (2024a) evaluate mathematical visual reasoning in MLLMs. These works conflate perception with reasoning, however, because they assess performance on complex tasks that combine diagram interpretation and mathematical reasoning. It thus remains unclear whether the performance truly reflects the models' ability to comprehensively understand the symbolic information in diagrams. Perceptual misinterpretations propagate downstream, leading to faulty reasoning and frequent hallucinations, although the final answer may occasionally still be correct Bai et al. (2024); Jiang et al. (2024); Wang et al. (2024a). Developing models capable of understanding symbolic information in diagrams is a critical milestone in advancing machine intelligence Cromley et al. (2010); de Rijke (1999).

Motivated by the above, we have designed a diagnostic benchmark to isolate and rigorously evaluate mathematical perception in MLLMs. MATHEMETRIC features problems that humans can solve "at a glance" without extensive reasoning. The benchmark contains 1,198 images and 1,609 carefully designed perception-oriented questions across four distinct task categories: shape classification, object counting, relationship identification, and object grounding. These tasks span three core mathematical domains—plane geometry, solid geometry, and graphical representations (line, bar, and pie graphs)—and question formats (multiple-choice, true/false, and free-form).

We systematically evaluate current MLLMs on MATHEMETRIC to investigate key questions:

***1. Do current MLLMs genuinely perceive mathematical diagrams?*** Fig. 1 presents the performance of eight current MLLMs—six generic models and two math-specific models—across four perception-focused tasks in MATHEMETRIC (see Tab. 1 for more MLLM results). Among the generic MLLMs, Qwen2.5-VL-7B Bai et al. (2025) achieves the highest average accuracy, followed by GPT-4o. But both suffer from severe hallucinations—responding to simple questions with unnecessarily long chains-of-thought and gener-

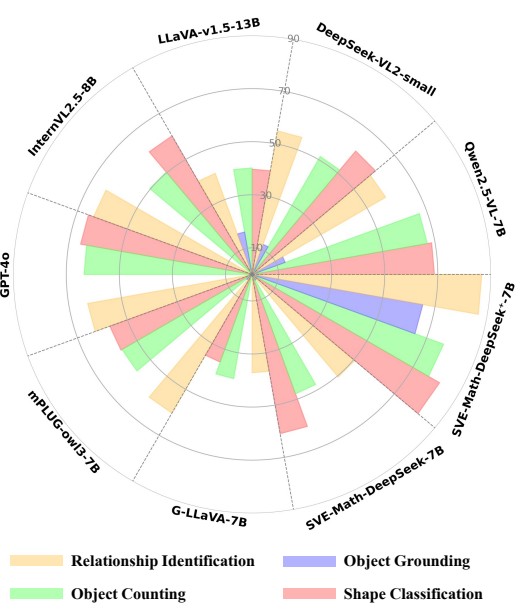

Figure 1: Performance on MATHEMETRIC reveals that diagram interpretation is challenging for MLLMs, particularly in fine-grained grounding tasks that require precise spatial localization. SVE-Math-DeepSeek[+]-7B, trained on GEOMETRIC, significantly outperforms comparators, validating the structure-aware geometric data design.

ating irrelevant visual content (see Fig. 6). The math-specific SVE-Math-DeepSeek Zhang et al. (2025), with a geometric primitive visual encoder, performs on par with Qwen2.5-VL-7B in shape classification and relationship identification. However, all models, including Qwen2.5-VL-7B and DeepSeek-VL2-Small, which are trained on nearly 2T natural images, consistently underperform on fine-grained bounding box grounding, with accuracy below 20%.

We then quantitatively analyze factors influencing MLLMs' perceptual capabilities in §3.3, providing more convincing evidence to answer this question. **Key findings are as follows:** (1) models are vulnerable to subtle visual noise and irrelevant distractors, failing to attend to salient objects; (2) when diagram–text conflicts arise, models over-rely on textual information, particularly those with weaker perceptual ability; and (3) models often resort to pattern memorization instead of perceptual understanding, as evidenced by their insensitivity to vertex ordering in shape classification, even though vertex order defines shape identity under formal geometric rules.

***2. Does stronger perceptual ability lead to better reasoning performance?*** To understand the potential reasons for weak diagram perception in MLLMs, we examine the limitations of existing diagram–caption training datasets, such as MAVIS Zhang et al. (2024b) and AutoGeo Huang et al. (2024), two of the largest in mathematical vision. They often contain ambiguous expressions and obscure structural properties of diagrams (Fig. 2a). As a means of investigation, we construct

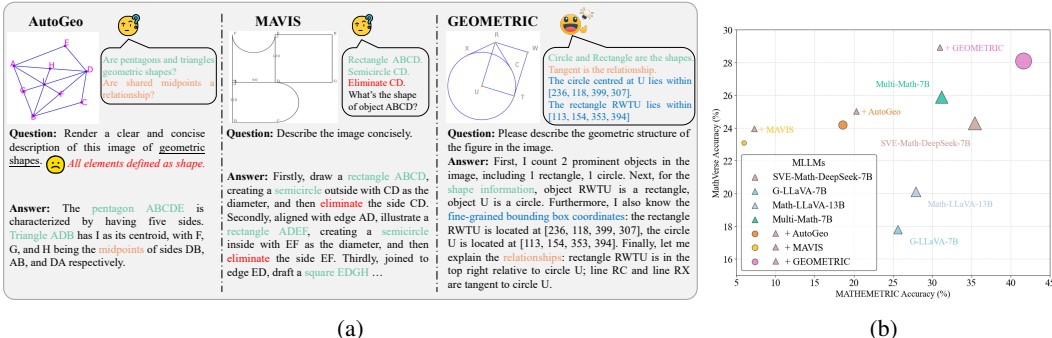

(a)                                                                (b)

Figure 2: Illustration of diagram-caption alignment training datasets, *w.r.t.*, AutoGeo, MAVIS, and our GEOMETRIC (Fig. 2a) with shapes in green, relationships in yellow, box locations in blue, and ambiguity in red. Fig. 2b demonstrates a positive correlation between low-level perception and high-level reasoning tasks, evaluated on MATHEMETRIC and MathVerse. Clear diagram perception leads to substantial improvements in mathematical reasoning performance.

GEOMETRIC, a high-quality {geometry-image, text} pairs that encode shapes, attributes, and interrelationships as graphs with fine-grained bounding box locations. Training on GEOMETRIC yields strong improvements in perception tasks, achieving +79% on the grounding task (Fig. 1). Additional results for Qwen2.5-VL-7B/32B trained on GEOMETRIC are presented in §3.2.

Moreover, in the close-up comparison with math-specific MLLMs (Fig. 2b, colored dot points), training with our proposed dataset substantially improves perceptual performance over AutoGeo and MAVIS, by 23.0% and 35.6%, respectively. In contrast, two alternative variants underperform the baseline (SVE-Math-DeepSeek-7B), with MAVIS diagram–caption pairs (despite being 5 × larger in scale than ours) yielding poorer results due to high uncertainty and the out-of-distribution nature of real-world geometric diagrams.

To further examine the impact on reasoning, we fine-tune those three models on the same reasoning dataset (MathV360K Shi et al. (2024)) for a fair comparison. **Two key observations emerge:** (1) our model achieves 28.1% accuracy on MathVerse, a +∼4% gain over others. This is notable because the improvement arises purely from enhanced perception, without additional reasoning data, whereas Multi-Math-7B Peng et al. (2024), trained with large-scale reasoning samples and reinforcement learning, achieves only 26.9%; (2) without clear visual guidance, models privilege verbal reasoning, termed blind reasoning. For example, with AutoGeo or MAVIS, models perform comparably to the base model on MathVerse under the same reasoning data, showing that structure-aware geometric samples provide essential visual understanding for accurate reasoning. Our model accurately identifies relevant visual elements, enabling it to generate more valid and faithful reasoning steps (see model responses in Figs. 7/16-18).

## 2 EVALUATION AND TRAINING SUITE DESIGN

### 2.1 MATHEMETRIC

Existing mathematical visual reasoning benchmarks often conflate perception with higher-level tasks such as numerical calculation and proof generation. As evaluation is based solely on final answers, intermediate perception errors remain hidden. Thus, it remains unclear whether MLLMs genuinely perceive diagrams or merely rely on the prior knowledge of powerful LLMs. We introduce MATHEMETRIC, the novel benchmark to evaluate MLLMs' perception-demanding abilities through both quantitative and qualitative analysis across coarse-to-fine granularity levels.

**Data Composition.** To comprehensively assess the perceptual abilities of MLLMs in mathematical contexts, our evaluation images cover plane geometry (66%), solid geometry (20%), and graphs (14%), including lines, bars, and pie charts. To facilitate MLLM evaluation, we formulate all tasks—except for bounding box grounding (free-form)—as multiple-choice or true/false question-answering problems. In total, we contribute 1,609 questions and 1,198 unique images, ensuring an even distribution across the different perception tasks. Detailed statistics for data composition are presented in Tab. 9 of the Appendix.

Figure 3: Sampled MATHEMETRIC examples from plane geometry *w.r.t.* each question-answer (Q&A) type, covering shape classification, object counting, relationship identification, and object grounding (from left to right). The green dotted bounding box is shown for illustration purposes only and are not provided as input to the models.

**Categorization.** The benchmark encompasses *shape classification*, *object counting*, *relationship identification*, and *object grounding*. Fig. 3 shows example illustrations of plane geometry (see §D for additional examples). Key features are as follows:

- Shape classification is a classic vision task where the model identifies object classes based on attributes, *i.e.*, vertices, material, color, and size. Our dataset includes a diverse set of geometric categories, comprising 16 basic shapes for plane geometry, 3 CLEVR-defined Johnson et al. (2017) objects for solid geometry, and 5 graphical elements as defined in FigureQA Kahou et al. (2017).

- Object counting requires models to determine either the total number of objects in an image or a specific shape count, *i.e.*, the number of circles or triangles present.

- Relationship identification evaluates models' understanding of 4 spatial and over 10 mathematical relationships between pairs of geometric primitives.

- Object grounding evaluates fine-grained localization by requiring MLLMs to accurately predict the top-left and bottom-right coordinates in the format (x1, y1, x2, y2) for an object within the image. This ensures that models can precisely identify and localize geometric structures based on textual descriptions.

**Question&Answer Construction.** Herein, we briefly introduce the construction of question–answer (Q&A) pairs for model evaluation. The detailed construction pipeline is presented in §F of the Appendix, which documents our synthetic data engine for generating plane geometry diagrams as well as the reformatted versions of the public CLEVR Johnson et al. (2017) and FigureQA Kahou et al. (2017) datasets for solid geometry and graph representations. This data engine enables controlled shape generation, relationship modeling, and visual attribute assignment, ensuring diverse and well-balanced datasets across perception tasks.

Fig. 4 describes the entire generation process: *1) Structured annotations for geometric primitives.* Inspired by AlphaGeometry Trinh et al. (2024), we use geometric clauses as fundamental units, combining basic shapes with mathematically valid interrelationships. Our synthetic data engine samples from a pool of 16 shapes (*e.g.*, isosceles triangle, square, rectangle, parallelogram, isosceles trapezoid, pentagon, circle . . . ellipse) and 10 relations (*e.g.*, parallel, perpendicular . . . tangent) and verifies logical consistency. The outputs are stored as structured JSON annotations specifying attributes, bounding boxes, and relationships, which are then used both for image rendering with Matplotlib Hunter (2007) and for generating question–answer (Q&A) pairs via template-based pipelines. *2) Q&A generation.* From structured JSON annotations, we employ a template-based pipeline to generate multiple-choice, true–false, and free-form questions for basic perception tasks. Correct answers are derived directly from annotations, while plausible distractors are generated to challenge geometric perception and combined with the correct answer into multiple-choice sets. For paired query diagrams, geometric clauses are translated into visual representations using Python code. To increase task difficulty, we apply image augmentations such as Gaussian noise, irregular scribbles, and wedge-shaped symbols for congruent angles or auxiliary lines (see Figs. 10 and 12 in the Appendix). *3) Reformatted public datasets.* To cover solid geometry and graphs, we reformat CLEVR Johnson et al. (2017) and FigureQA Kahou et al. (2017) into the same structured format, enabling consistent Q&A generation across domains.

After generating our synthetic data and collecting public datasets, we conduct a comprehensive review to verify answer accuracy, ensure consistency between questions and diagrams, and confirm relevance to the four perception tasks, ensuring high-quality and precise dataset annotations.

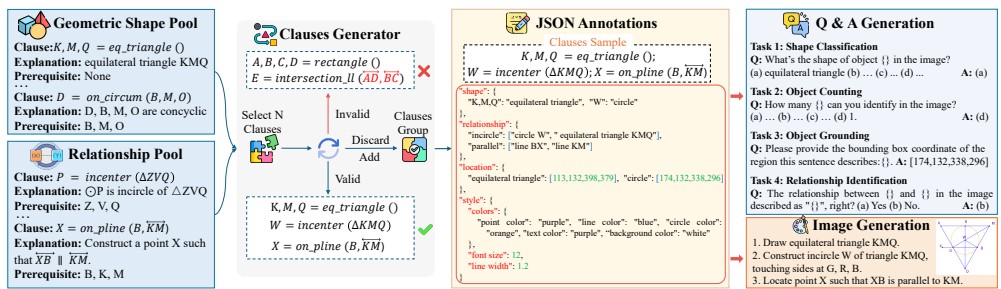

Figure 4: We synthesize geometric figures by randomly sampling elements from the geometric shape pool and relationship pool, ensuring consistency through a verifier that enforces logical constraints based on manually designed rules, fundamental mathematical principles, and prerequisite points. All visual elements are structured and saved in JSON format. Images are rendered using the Matplotlib package, and corresponding Q&A pairs are generated using a template-based pipeline.

## 2.2 PERCEPTION-ORIENTED TRAINING DATASET (GEOMETRIC)

Evaluation results on MATHEMETRIC reveal that both open-source and closed-source MLLMs struggle to identify relevant visual regions in symbolic and abstract mathematical diagrams, despite their strong performance in theoretical reasoning and numerical computation. This contrasts with human cognitive abilities, where low-level perceptual tasks are typically solved rapidly compared to high-level reasoning tasks. This raises an important question: why do generic MLLMs—despite being trained on large-scale visual datasets—struggle to perceive mathematical diagrams? The key reason lies in the domain gap between natural images and mathematical diagrams Lu et al.; Zhang et al. (2025). Unlike semantically rich natural images (bitmaps), diagrams are defined by precise geometric structures and symbolic notations (vectors). Without superficial patterns or semantic priors to exploit, MLLMs fail to generalize Geirhos et al. (2022). This underscores the need for datasets that reflect the uniquely structured, graph-like nature of diagrams. Moreover, learning from explicit structures within the data context would reduce learning complexity and enhance the problem-solving abilities of models Han et al. (2025).

We thus develop a structured visual dataset—where the training corpus is built from object attributes and extends to other objects based on their relationships—to improve visual attention and mitigate MLLMs' reliance on textual shortcuts. In specifc, GEOMETRIC follows a structured format:

First, I count $\{N\}$ prominent object(s) in the image. Next, for shape information, object $\{attrbu.^i\}$ is a $\{shape^i\}$, and $\cdots$. Furthermore, I also know the fine-grained bounding box coordinates: the $\{shape^i\}$ $\{attrbu.^i\}$ is located at $\{box\_cor.^i\}$, and $\cdots$. Finally, let me explain the relationships: the $\{shape^i\}$ $\{attrbu.^i\}$ $\{rela.^{ij}\}$ to the $\{shape^j\}$ $\{attrbu.^j\}$, and $\cdots$.

To further enhance the model's ability to follow instructions, we construct a task-specific instruction dataset in a multi-turn conversation format. Each question is tailored to a specific perception task, with answers presented either in free-form or as a selected option. For example: **Q**: What is the shape of object $\{attrbu.^i\}$? **A**: $\{shape^i\}$. See §G in the Appendix for additional demonstrations.

Overall, GEOMETRIC contributes to MLLM training by: (1) providing clear object attributes and their relationships, akin to graph nodes and edges, in training QA pairs; (2) offering fine-grained bounding box coordinates of elements, enabling models to systematically learn spatial awareness; and (3) integrating with reasoning-based CoT mathematical visual datasets during self-fine-tuning stage, allowing models to both perceive and reason accurately.

## 3 EXPERIMENTS

### 3.1 EXPERIMENTAL SETUP

**Generic and Mathematical MLLMs.** We evaluate 20 MLLMs on MATHEMETRIC across plane geometry, solid geometry, and graphs, including closed-source generic models such as GPT-4o and GPT-o1, as well as open-source models like LLaVA-v1.5 Liu et al. (2023b), mPLUG-Owl3 Ye et al. (2024), InternLM-XComposer2 Dong et al. (2024), Qwen2VL Wang et al. (2024c), Qwen2.5VL Bai et al. (2025), DeepSeek-VL2 Wu et al. (2024), InternVL2 Chen et al. (2024b), InternVL2.5 Chen

Table 1: Performance comparison of different MLLMs on MATHEMETRIC across plane geometry, solid geometry, and graphs. *cls*, *cnt*, *grd*, and *rlat* represent different question categories: shape classification, object counting, object grounding, and relationship identification, respectively. *all* indicates the overall accuracy, calculated as the ratio of correctly answered questions to the total number of questions in the benchmark, while **Avg.** denotes the average *all* score across all subjects.

| Model | Size | Avg. | Plane Geometry | | | | | Solid Geometry | | | | | Graphs | | | | |
|---|---|---|---|---|---|---|---|---|---|---|---|---|---|---|---|---|---|
| | | | all | cls | cnt | grd | rlat | all | cls | cnt | grd | rlat | all | cls | cnt | grd | rlat |
| *Human* | | | | | | | | | | | | | | | | | |
| Authors (ours) | – | 99.2 | 98.5 | 98.7 | 99.3 | 95.9 | 100.0 | 99.3 | 100.0 | 100.0 | 98.7 | 98.3 | 99.8 | 100.0 | 100.0 | 99.3 | 100.0 |
| *Open-Source Generic MLLMs* | | | | | | | | | | | | | | | | | |
| LLaVA-v1.5 Liu et al. (2023b) | 7B | 33.3 | 29.2 | 29.0 | 39.6 | 14.2 | 37.5 | 31.6 | 43.0 | 42.3 | 0.0 | 31.3 | 39.0 | 76.8 | 35.2 | 0.0 | 39.4 |
| LLaVA-v1.5 Liu et al. (2023b) | 13B | 35.4 | 32.8 | 29.3 | 40.4 | 23.5 | 42.0 | 35.9 | 60.5 | 38.1 | 0.0 | 35.0 | 37.6 | 63.8 | 42.6 | 0.0 | 45.5 |
| mPLUG-Owl3 Ye et al. (2024) | 7B | 50.0 | 36.4 | 46.7 | 41.6 | 3.9 | 58.5 | 65.3 | 95.4 | 83.5 | 0.0 | 62.5 | 48.2 | 59.4 | 77.8 | 0.0 | 66.7 |
| InternLM-XComposer2 Dong et al. (2024) | 7B | 55.6 | 35.8 | 49.4 | 48.8 | 0.0 | 47.0 | 62.9 | 90.7 | 86.6 | 0.0 | 53.8 | 54.6 | 60.9 | 94.4 | 0.0 | 78.8 |
| Qwen2-VL Wang et al. (2024c) | 7B | 51.4 | 37.9 | 47.6 | 41.2 | 12.8 | 53.0 | 64.1 | 93.0 | 78.4 | 14.3 | 55.0 | 52.3 | 84.1 | 88.9 | 3.2 | 18.2 |
| Qwen2-VL Wang et al. (2024c) | 72B | 59.9 | 42.4 | 51.2 | 50.8 | 17.4 | 52.0 | 71.2 | 97.7 | 84.5 | 6.4 | 77.5 | 66.1 | 76.8 | 98.2 | 16.1 | 84.9 |
| Qwen2.5-VL Bai et al. (2025) | 7B | 59.2 | 44.0 | 56.2 | 51.3 | 18.5 | 52.0 | 68.0 | 98.8 | 88.7 | 0.0 | 65.0 | 65.7 | 89.9 | 100.0 | 3.2 | 78.8 |
| Qwen2.5-VL Bai et al. (2025) | 32B | 62.2 | 43.3 | 56.9 | 54.8 | 0.0 | 67.0 | 72.5 | 98.8 | 89.7 | 1.6 | 87.5 | 68.8 | 91.3 | 100.0 | 1.6 | 97.0 |
| DeepSeek-VL2-Tiny Wu et al. (2024) | 3B | 32.6 | 29.5 | 45.2 | 34.4 | 4.6 | 32.0 | 39.0 | 76.7 | 32.0 | 0.0 | 37.5 | 29.4 | 39.1 | 57.4 | 0.0 | 18.2 |
| DeepSeek-VL2-Small Wu et al. (2024) | 16B | 51.5 | 37.6 | 47.6 | 43.6 | 12.5 | 48.5 | 63.8 | 98.8 | 70.1 | 11.1 | 60.0 | 53.2 | 76.8 | 53.7 | 11.3 | 81.8 |
| InternVL2 Chen et al. (2024b) | 8B | 48.4 | 31.9 | 44.3 | 38.0 | 0.0 | 48.5 | 62.9 | 98.8 | 62.9 | 4.8 | 70.0 | 50.5 | 68.1 | 75.9 | 0.0 | 66.7 |
| InternVL2.5 Chen et al. (2024a) | 8B | 50.7 | 35.0 | 48.8 | 36.0 | 0.0 | 60.0 | 65.6 | 98.8 | 72.2 | 4.8 | 70.0 | 51.4 | 68.1 | 77.8 | 0.0 | 69.7 |
| InternVL2.5 Chen et al. (2024a) | 38B | 63.1 | 44.0 | 59.9 | 52.0 | 2.5 | 66.0 | 78.8 | 98.8 | 92.8 | 38.1 | 72.5 | 66.5 | 98.6 | 96.3 | 3.2 | 69.7 |
| Vision-R1 Huang et al. (2025) | 7B | 58.2 | 39.6 | 53.9 | 45.6 | 0.0 | 64.0 | 66.6 | 95.4 | 89.7 | 0.0 | 60.0 | 68.4 | 97.1 | 100.0 | 0.0 | 84.9 |
| MINT-CoT Chen et al. (2025) | 7B | 54.7 | 39.1 | 52.7 | 46.4 | 2.9 | 58.0 | 61.7 | 96.5 | 67.0 | 6.4 | 61.3 | 63.3 | 82.6 | 92.6 | 0.0 | 93.9 |
| **Qwen2.5-VL⁺ (ours)** | 7B | 72.9 | 78.5 | 70.7 | 79.2 | 82.6 | 85.0 | 71.9 | 97.9 | 86.2 | 12.9 | 70.0 | 68.2 | 94.2 | 96.3 | 4.9 | 89.4 |
| **Qwen2.5-VL⁺ (ours)** | 32B | 74.2 | 77.9 | 70.7 | 79.6 | 84.0 | 79.5 | 73.8 | 98.8 | 86.4 | 15.0 | 85.0 | 71.1 | 98.6 | 98.2 | 2.7 | 99.0 |
| *Closed-Source Generic MLLMs* | | | | | | | | | | | | | | | | | |
| GPT-4o | - | 53.3 | 42.8 | 58.4 | 53.2 | 1.1 | 62.5 | 60.7 | 72.1 | 84.5 | 1.6 | 66.3 | 56.4 | 92.8 | 72.2 | 1.6 | 57.6 |
| GPT-o1 | - | 36.5 | 15.8 | 33.2 | 11.6 | 0.0 | 14.0 | 41.4 | 75.6 | 52.6 | 0.0 | 23.8 | 52.3 | 82.6 | 81.5 | 0.0 | 39.4 |
| GPT-o4-mini-high | - | 48.0 | 19.1 | 29.3 | 24.4 | 0.4 | 21.5 | 64.7 | 94.2 | 79.4 | 0.0 | 66.3 | 60.1 | 95.7 | 77.8 | 0.0 | 69.7 |
| *Open-Source Mathematical MLLMs* | | | | | | | | | | | | | | | | | |
| Math-LLaVA Shi et al. (2024) | 13B | 40.0 | 27.9 | 34.4 | 32.4 | 0.0 | 50.5 | 44.8 | 81.4 | 55.7 | 0.0 | 27.5 | 47.3 | 78.3 | 59.3 | 0.0 | 51.5 |
| G-LLaVA Gao et al. (2023) | 7B | 30.3 | 25.6 | 27.8 | 41.2 | 0.4 | 38.0 | 32.3 | 45.4 | 38.1 | 4.8 | 32.5 | 33.9 | 58.0 | 37.0 | 0.0 | 42.4 |
| Math-PUMA-DeepSeek-Math-VL Zhuang et al. (2025) | 7B | 44.7 | 29.3 | 45.8 | 26.8 | 0.0 | 46.0 | 46.9 | 76.7 | 62.9 | 0.0 | 32.5 | 57.8 | 92.8 | 75.9 | 0.0 | 63.6 |
| URSA Luo et al. (2025) | 8B | 42.2 | 31.8 | 39.2 | 39.2 | 0.0 | 55.0 | 40.2 | 79.1 | 41.2 | 0.0 | 28.8 | 54.6 | 82.6 | 68.5 | 0.0 | 75.8 |
| MultiMath Peng et al. (2024) | 7B | 42.1 | 31.2 | 44.0 | 30.4 | 1.1 | 53.0 | 46.7 | 81.4 | 53.6 | 4.7 | 33.8 | 48.6 | 79.7 | 57.4 | 3.2 | 33.8 |
| SVE-Math-DeepSeek Zhang et al. (2025) | 7B | 46.6 | 35.4 | 52.4 | 36.0 | 3.6 | 51.0 | 49.4 | 77.9 | 62.9 | 1.5 | 41.3 | 55.1 | 81.2 | 75.9 | 1.6 | 69.7 |
| **SVE-Math-DeepSeek⁺ (ours)** | 7B | 68.4 | 84.6 | 75.8 | 88.4 | 82.9 | 96.5 | 54.1 | 85.3 | 65.8 | 20.3 | 45.0 | 60.7 | 85.1 | 78.4 | 1.6 | 75.7 |

et al. (2024a), Vision-R1 Huang et al. (2025) and MINT-CoT Chen et al. (2025). Additionally, we assess math-specific MLLMs, including SVE-Math-DeepSeek Zhang et al. (2025), Math-LLaVA Shi et al. (2024), G-LLaVA Gao et al. (2023), URSA Luo et al. (2025) and MultiMath Peng et al. (2024). This comprehensive evaluation provides insights into the diagram perception capabilities of state-of-the-art multimodal models.

**Implementation Details.** For open- and closed-source generic MLLMs, we follow official inference settings, including temperature, number of beams, and maximum token length. For open-source mathematical LLMs, we adopt the standard configurations from SVE-Math-DeepSeek, setting the temperature to 0, the number of beams to 1, and the maximum token length to 1024. Choices are extracted using predefined rules tailored to each MLLM's output format. We apply GEOMETRIC for full-parameter SFT on SVE-Math-DeepSeek and parameter-efficient LoRA training on Qwen2.5-VL-7B/32B. See §C for training implementation details.

## 3.2 MAIN RESULTS

Tab. 1 summarizes the performance of current MLLMs on plane geometry, solid geometry Johnson et al. (2017), and graphs Kahou et al. (2017). Below, we provide an analysis of their performance on MATHEMETRIC.

**Generic MLLMs.** General-purpose models trained on diverse datasets, including tables, charts, and documents Chen et al. (2019); Kahou et al. (2017); Yuan et al. (2022), as well as visual grounding datasets Shao et al. (2019); You et al. (2024), still perform poorly on mathematical diagram perception. In particular, their performance in plane geometry remains low, with most models scoring below 45% on average. For solid geometry and graphs, general-purpose models significantly outperform mathematical MLLMs, due to their exposure to large-scale FigureQA Kahou et al. (2017), CLEVR Johnson et al. (2017), and various chart understanding datasets Yuan et al. (2022). However, they still fail in fine-grained box-level tasks, with most models achieving 0 accuracy at an IoU threshold of 0.65, and lag significantly behind human-level perception. The symbolic and structured diagrams require genuine visual understanding rather than superficial pattern recognition. Without this, models fail to identify where to look, leading to poor perception performance. Notably, scaling

up model size is neither an optimal nor an effective solution, as it provides only marginal gains in perception compared to reasoning benchmarks. For instance, increasing Qwen2VL from 7B to 72B improves top-1 accuracy by 22.3% on MathVista but only 8.3% on MATHEMETRIC.

**Mathematical MLLMs.** Models such as MultiMath, Math-LLaVA, and SVE-Math-DeepSeek, fine-tuned from generic MLLMs (*i.e.*, LLaVA and DeepSeek) using mathematical visual data, achieve strong reasoning performance on MathVerse Zhang et al. (2024a), MathVista Lu et al., GeoQA Gao et al. (2023) and MATH-V Wang et al. (2024b), but struggle with geometric perception across all three subjects.

Among them, SVE-Math-DeepSeek outperforms other mathematical MLLMs by incorporating a primitive visual encoder trained with detection and boundary segmentation losses. As shown in Tab. 2, SVE-Math-DeepSeek$^+$ is an enhanced version trained on GEOMETRIC, and it achieves significant gains in plane geometry perception. Notably, even without direct training on solid geometry

Table 2: Performance comparison on math perception and reasoning benchmarks.

| Model | MATHEMETRIC | | | MathVerse | MathVista | GeoQA | MATH-V |
|---|---|---|---|---|---|---|---|
| | Plane | Soild | Graphs | | | | |
| Qwen2.5-VL-7B Bai et al. (2025) | 44.0 | 69.0 | 65.7 | 49.2 | 68.2 | 76.4 | 25.1 |
| Vision-R1-7B Huang et al. (2025) | 39.6 | 66.6 | 68.4 | 52.4 | 73.5 | 78.9 | - |
| MINT-CoT-7B-SFT Chen et al. (2025) | 39.1 | 61.7 | 63.3 | - | 67.8 | 62.1 | - |
| Qwen2.5-VL-32B Bai et al. (2025) | 43.3 | 72.5 | 68.8 | 54.8 | 74.7 | 82.9 | 31.9 |
| Math-LLaVA-13B Shi et al. (2024) | 27.9 | 44.8 | 47.3 | 20.1 | 46.6 | 60.7 | 15.7 |
| G-LLaVA-7B Gao et al. (2023) | 25.6 | 31.3 | 33.9 | 17.8 | 25.6 | 64.2 | 12.1 |
| MultiMath-7B Peng et al. (2024) | 31.2 | 45.7 | 48.6 | 25.9 | 49.3 | 74.1 | - |
| SVE-Math-DeepSeek-7B Zhang et al. (2025) | 35.4 | 49.4 | 55.1 | 24.3 | 48.7 | 72.8 | 14.4 |
| **SVE-Math-DeepSeek-7B$^+$ (ours)** | **84.6** | 54.1 | 60.7 | 28.1 | 51.3 | 76.2 | 16.6 |
| **Qwen2.5-VL-7B$^+$ (ours)** | 78.5 | 71.9 | 68.2 | 52.8 | 70.3 | 79.6 | 27.3 |
| **Qwen2.5-VL-32B$^+$ (ours)** | 77.9 | **74.2** | **71.1** | **57.3** | **76.9** | **85.3** | **33.3** |

and graphs, it outperforms others due to its improved ability to discriminate relationships and understand spatial configurations. To further validate the positive correlation between perception and reasoning, we fine-tune strong generic Qwen2.5-VL-7B/32B on GEOMETRIC. Tab. 2 shows that our dataset yields $\sim +4\%$ improvements over SVE-Math-DeepSeek and $\sim +3\%$ gains over Qwen2.5-VL-7B/32B across four mathematical reasoning benchmarks. Although we do not employ explicit visual–text integration mechanisms, as done in MINT-CoT Chen et al. (2025), our model still exhibits automatic adaptation and task transfer from low-level perception to high-level reasoning, demonstrating that these two abilities are complementary. As shown in the qualitative comparisons between the base model and the GEOMETRIC-fine-tuned model (Figs. 7/16–18), many previously incorrect cases are resolved simply by correcting a key perceptual error, and the resulting improved perception stabilizes multi-step reasoning chains. Another related line of work is reinforcement-learning–based reasoning enhancement, as in Vision-R1 Huang et al. (2025), which incentivizes reasoning ability built on large-scale and corss-domain perception–reasoning datasets, requiring much heavier computational resources. Exploring how to explicitly model visual–text interactions remains a promising direction for understanding how enhanced perception can further support and strengthen reasoning. We provide an in-depth analysis in § C.

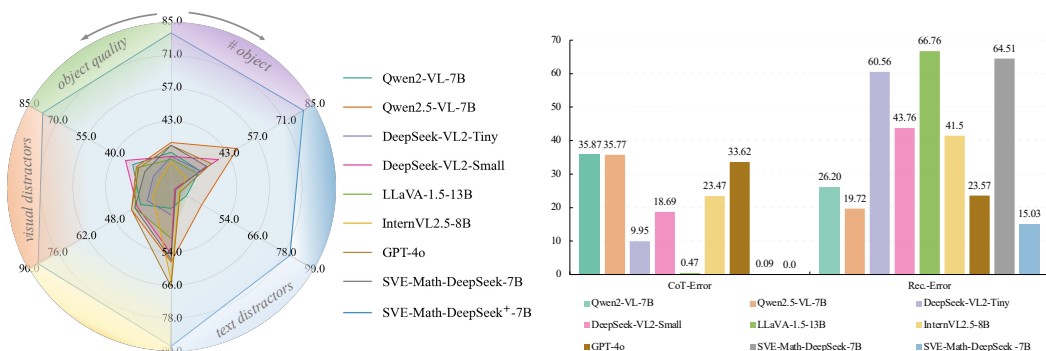

Figure 5: We evaluate five key factors influencing MLLM perception: object count (# obejct), visual quality, visual distractors, textual distractors, and Chain-of-Thought (CoT) reasoning, with close-up results shown in Tabs 3-4 and Fig. 6.

Figure 6: Perceptual errors under Chain-of-Thought (CoT) *vs.* direct reasoning. Chain-of-thought (CoT) errors occur when a model provides step-by-step explanations yet yields an incorrect answer.

## 3.3 ABLATION STUDY

Since mathematical models are trained exclusively on math-domain datasets, they offer a more controlled setting for analyzing perception capabilities compared to general-purpose MLLMs. Therefore, we choose SVE-Math-DeepSeek as the base model for the below ablation study.

### 3.3.1 KEY FACTORS INFLUENCING DIAGRAM PERCEPTION

We examine five key factors: the number of objects, object quality, visual distractors, text distractors, and Chain-of-Thought (CoT) responses. While those factors are by no means exhaustive, they aim to illuminate some of the fundamental perceptual limitations of current MLLMs, providing insights for practical applications and future improvements. The overall accuracy (*all*) on plane geometry from MATHEMETRIC across nine MLLMs is presented in Fig. 5.

**The Number of Objects.** Visual perception complexity increases with the number of objects. We observe that as the number of geometric shapes grows, models struggle with accurate object localization, shape classification, and counting. When scaling the number of objects from 1 to $N$ ($N < 5$), Qwen-series model's performance drops by approximately 12%, the DeepSeek-series by 18%, and GPT-4o by 15%. Our method and its baseline show declines of 6% and 13%.

Table 3: Accuracy (%) on plane geometry perception *w.r.t.* object quality (w/ or w/o Gaussian noise) and visual distractor injection.

| MATHEMETRIC | Noise $\sim \mathcal{N}(0,0.3)$ w/o | w/$_{\Delta\uparrow\downarrow}$ | Distractors w/o | w/$_{\Delta\uparrow\downarrow}$ |
|---|---|---|---|---|
| Qwen2VL-7B | 29.8 | $33.9_{\uparrow4.1}$ | 29.8 | $26.2_{\downarrow3.6}$ |
| Qwen2.5VL-7B | 33.9 | $32.1_{\downarrow1.8}$ | 33.9 | $31.0_{\downarrow2.9}$ |
| DeepSeek-VL2-Tiny | 27.9 | $23.8_{\downarrow4.1}$ | 27.9 | $22.6_{\downarrow5.3}$ |
| DeepSeek-VL2-Small | 27.9 | $37.5_{\uparrow9.6}$ | 27.9 | $25.9_{\downarrow2.0}$ |
| LLaVA-v1.5-13B | 26.8 | $31.6_{\uparrow4.8}$ | 26.8 | $29.1_{\uparrow2.3}$ |
| InternVL2.5-8B | 25.6 | $20.8_{\downarrow4.8}$ | 25.6 | $19.6_{\downarrow6.0}$ |
| GPT-4o | 32.7 | $30.9_{\downarrow1.8}$ | 32.7 | $28.6_{\downarrow4.1}$ |
| SVE-Math-DeepSeek-7B | 32.8 | $28.0_{\downarrow4.8}$ | 32.8 | $29.2_{\downarrow3.6}$ |
| **SVE-Math-DeepSeek$^+$-7B** | 80.7 | $78.3_{\downarrow2.4}$ | 80.7 | $80.3_{\downarrow0.4}$ |

Table 4: Performance of relationship identification (*rlat*) on MATHEMETRIC under different textual distractor settings.

| MATHEMETRIC (*rlat*) | Unrela. infor. w/o | w/$_{\Delta\uparrow\downarrow}$ | Conflicts w/o | w/$_{\Delta\uparrow\downarrow}$ |
|---|---|---|---|---|
| Qwen2VL-7B | 53.0 | $48.5_{\downarrow4.5}$ | 53.0 | $24.5_{\downarrow28.5}$ |
| Qwen2.5VL-7B | 52.0 | $56.0_{\uparrow4.0}$ | 52.0 | $30.0_{\downarrow22.0}$ |
| DeepSeek-VL2-Tiny | 32.0 | $34.5_{\uparrow2.5}$ | 32.0 | $3.5_{\downarrow28.5}$ |
| DeepSeek-VL2-Small | 48.5 | $42.5_{\downarrow6.0}$ | 48.5 | $20.5_{\downarrow28.0}$ |
| LLaVA-v1.5-13B | 42.0 | $37.5_{\downarrow4.5}$ | 42.0 | $16.0_{\downarrow26.0}$ |
| InternVL2.5-8B | 60.0 | $49.0_{\downarrow11.0}$ | 60.0 | $18.0_{\downarrow42.0}$ |
| GPT-4o | 62.5 | $63.5_{\uparrow1.0}$ | 62.5 | $38.0_{\downarrow24.5}$ |
| SVE-Math-DeepSeek-7B | 51.0 | $49.0_{\downarrow2.0}$ | 51.0 | $15.0_{\downarrow36.0}$ |
| **SVE-Math-DeepSeek$^+$-7B** | 96.5 | $95.0_{\downarrow1.5}$ | 96.5 | $82.5_{\downarrow14.0}$ |

Table 5: Additional results for "blind faith in text" phenomenon. Accuracy (%) on plane-geometry (*rlat*) under explicit modality-priority prompts.

| MATHEMETRIC (*rlat*) | Conflicts w/o | Neutral w/$_{\Delta\uparrow\downarrow}$ | Visual-priority w/$_{\Delta\uparrow\downarrow}$ | Text-priority w/$_{\Delta\uparrow\downarrow}$ |
|---|---|---|---|---|
| Qwen2VL-7B | 53.0 | $24.5_{\downarrow28.5}$ | $25.5_{\downarrow27.5}$ | $25.5_{\downarrow27.5}$ |
| Qwen2.5VL-7B | 52.0 | $30.0_{\downarrow22.0}$ | $30.5_{\downarrow21.5}$ | $28.3_{\downarrow23.7}$ |
| DeepSeek-VL2-Tiny | 32.0 | $3.5_{\downarrow28.5}$ | $11.0_{\downarrow21.0}$ | $9.5_{\downarrow22.5}$ |
| DeepSeek-VL2-Small | 48.5 | $20.5_{\downarrow28.0}$ | $22.0_{\downarrow26.5}$ | $19.5_{\downarrow29.0}$ |
| LLaVA-v1.5-13B | 42.0 | $16.0_{\downarrow26.0}$ | $15.0_{\downarrow27.0}$ | $14.5_{\downarrow27.5}$ |
| InternVL2.5-8B | 60.0 | $18.0_{\downarrow42.0}$ | $17.0_{\downarrow43.0}$ | $15.5_{\downarrow44.5}$ |
| GPT-4o | 62.5 | $38.0_{\downarrow24.5}$ | $39.0_{\downarrow23.5}$ | $34.0_{\downarrow28.5}$ |
| SVE-Math-DeepSeek-7B | 51.0 | $15.0_{\downarrow36.0}$ | $15.0_{\downarrow36.0}$ | $12.5_{\downarrow38.5}$ |
| **SVE-Math-DeepSeek$^+$-7B** | 96.5 | $82.5_{\downarrow14.0}$ | $87.5_{\downarrow9.0}$ | $84.0_{\downarrow12.5}$ |

Table 6: Accuracy (%) on plane geometry for shape classification (*cls*) w.r.t. vertex ordering (clockwise *vs.* random).

| MATHEMETRIC (*cls*) | Random | Clockwise |
|---|---|---|
| Qwen2VL-7B | 47.4 | 47.6 |
| Qwen2.5VL-7B | 56.7 | 56.2 |
| DeepSeek-VL2-Tiny | 45.0 | 45.2 |
| DeepSeek-VL2-Small | 47.1 | 47.6 |
| LLaVA-v1.5-7B | 29.8 | 29.0 |
| InternVL2.5-8B | 48.3 | 48.8 |
| GPT-4o | 57.8 | 58.4 |
| SVE-Math-DeepSeek-7B | 51.2 | 52.4 |
| **SVE-Math-DeepSeek$^+$-7B** | **74.9** | **75.8** |

**Visual Distractors.** We draw visual distractors—irregular scribbles, wedge-shaped angle markers, and auxiliary lines—to evaluate MLLMs' ability to focus on relevant geometric elements (see Fig. 10). Despite explicitly prompting models to ignore distractors (*e.g.*, the input diagram contains visual distractors on the target foreground objects; please ignore them when performing the perception tasks), most are still negatively affected. Close-up results are shown in Tab. 3. All evaluated MLLMs show a performance drop of 2–6% under visual distractors, except LLaVA-1.5-13B, which improves by 2.3%. Our model remains robust, showing minimal sensitivity to distractors and better visual focus on geometric primitives.

**Object Quality.** For visual fidelity analysis, we apply Gaussian noise to degrade object quality. As shown in Tab. 3, with a variance of 0.3, most models degrade, while Qwen2VL, DeepSeek-VL2-Small, and LLaVA-1.5-13B unexpectedly improve. This may result from exposure to degraded visuals during training, making noisy images better aligned with their learned distribution—especially given the lack of clean geometric diagrams in the training data. To enable a more comprehensive

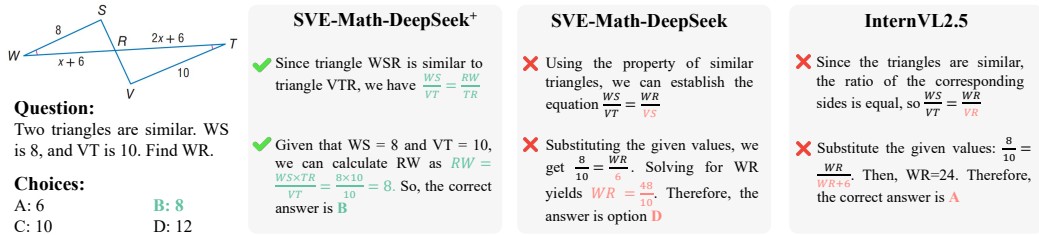

Figure 7: Response comparisons between SVE-Math-DeepSeek[+]-7B, SVE-Math-DeepSeek-7B, and InterVL2.5-8B in MathVerse. More demos are shown in Figs. 16-18 of the Appendix.

evaluation and mitigate data bias, we introduce stronger noise (variance 0.5/0.8). As distortion increases, all models show sharp drops in geometric recognition. Accuracy under noise is shown in Fig. 11, with example distortions in Fig. 12.

**Text Distractors.** To examine the influence of textual modality in vision-centered tasks, we evaluate two settings within the relationship identification task. The first introduces irrelevant information unrelated to the target, such as foreground/background colors or object vertices. The second presents contradictory cues: a diagram with clear visual cues (*e.g.*, two visibly parallel lines) and no ambiguous text; and the same diagram but with contradictory textual distractors added during inference (*e.g.*, the two lines are perpendicular). In both cases, correctness is determined by whether the model's answer matches the ground truth defined by the visual cues in the diagram, irrespective of the textual guidance. The results in Tab. 4 show that providing conflicting knowledge significantly impairs the perceptual ability of all evaluated MLLMs. The impact is especially pronounced in models with weaker perception capabilities such as InternVL2.5-8B (48.4% in **Avg.**), while stronger perceptual models such as Qwen2.5-VL-7B (59.2% in **Avg.**) are less affected. Our method, which achieves the best perceptual performance, demonstrates greater robustness—but still suffers a 14% drop. This highlights that MLLMs tend to over-rely on textual input when inconsistencies arise, exhibiting a "blind faith in text".

These conflict experiments were conducted under neutral settings, where no modality was favored during inference. To further examine whether explicit prompting can mitigate this behavior, we conducted additional controlled experiments using system prompts that explicitly modulate modality priority: *visual-priority prompt*: "Carefully examine the diagram and prioritize visual information. Only use text labels to confirm what you observe visually"; *text-priority prompt*: "Focus on the textual labels and annotations in the diagram. Use the visual structure to support the textual information". The results in Tab. 5 show that even when explicitly instructed to prioritize visual information, the model still exhibits "blind faith in text", although performance is marginally higher than under the text-priority prompt. Addressing this limitation should therefore be an important focus for future research.

**Chain-of-Thought (CoT) Response.** CoT reasoning is designed to enhance step-by-step logical inference, yet its effectiveness in visual perception tasks remains uncertain. Our analysis shows that while CoT improves textual reasoning, it does not directly enhance spatial or geometric understanding. Models incorporating CoT reasoning often struggle with fundamental perception tasks, leading to significant CoT errors. Models often generate excessive yet irrelevant rationale misaligned with the diagram, ultimately resulting in incorrect responses, particularly in Qwen-series models and GPT-4o (over 30% in Fig. 6). See §I for CoT response examples in the Appendix.

### 3.3.2 DIAGRAM UNDERSTANDING REMAINS CHALLENGING

To further examine the diagram understanding of MLLMs, we design a vertex-ordering ablation. Vertex order is a strict mathematical constraint: in a consistent clockwise order, vertices must form a closed shape. This ablation tests whether MLLMs rely on genuine geometric understanding or merely statistical pattern matching. We therefore randomize vertex order for each object, and results in Tab. 6 show that models are insensitive to this change. This insensitivity suggests that MLLMs do not internalize the geometric rules governing shape formation, but instead depend on surface-level patterns. Enabling models to grasp such geometric constraints remains a challenging yet promising direction for advancing true diagram understanding.

### 3.3.3 COUNTING OVERLAPPING OBJECTS IS OVERLY DEMANDING

In our synthetic plane-geometry diagrams, we preserve object separations to reduce ambiguity in counting, but overlaps remain unavoidable as object density increases. By default, MATHEMETRIC directs models to count only *prominent* objects, ignoring complex overlaps. We ablate this design choice by requiring models to count all potential overlaps. Results show that overlapping cases are far more difficult than non-overlapping ones (54.8 *v.s.* 32.8 for Qwen2.5-VL-7B; 88.2 *v.s.* 67.3 for SVE-Math-DeepSeek $^+$-7B). See Tab. 10 and §C for detailed analysis and settings.

Table 7: Performance comparison *w.r.t.* variants trained on GEOMETRIC and other mathematical alignment datasets. ★ is the pretrained model continuously self-fine-tuned (SFT) using MathV360K and instruction-formatted GEOMETRIC.

| Model | | MATHEMETRIC | | MathVerse | MathVista |
|---|---|---|---|---|---|
| | Plane | Soild | Graphs | | |
| SVE-Math-DeepSeek | 35.4 | 49.4 | 55.1 | 24.3 | 48.7 |
| Align. +AutoGeo (△) | 18.6 | 34.7 | 40.4 | 17.8 | 45.0 |
| +MAVIS (◇) | 6.0 | 2.5 | 6.9 | 6.8 | 34.3 |
| +GEOMETRIC (♡) | 41.6 | 40.2 | 49.2 | 19.7 | 46.2 |
| SFT △ • ★ | 79.6 | 44.2 | 55.1 | 25.2 | 48.3 |
| ◇ • ★ | 78.1 | 42.3 | 54.1 | 23.3 | 47.1 |
| ♡ • ★ | 84.6 | 54.1 | 60.7 | 28.1 | 51.3 |

Table 8: Performance comparison on math perception and reasoning benchmarks. The symbols ● and ○ denote models trained without GEOMETRIC and with a frozen visual encoder during the SFT, respectively.

| Model | MATHEMETRIC | | | MathVerse | MathVista | GeoQA |
|---|---|---|---|---|---|---|
| | Plane | Soild | Graphs | | | |
| SVE-Math-DeepSeek-7B | 35.4 | 49.4 | 55.1 | 24.3 | 48.7 | 72.8 |
| SVE-Math-DeepSeek$^+$-7B | 84.6 | 54.1 | 60.7 | 28.1 | 51.3 | 76.2 |
| SVE-Math-DeepSeek$^+$-7B (●) | 35.4 | 48.0 | 54.3 | 25.0 | 49.1 | 72.5 |
| SVE-Math-DeepSeek$^+$-7B (○) | 82.9 | 53.6 | 61.2 | 26.2 | 50.1 | 74.6 |
| Qwen2.5-VL-7B | 44.0 | 69.0 | 65.7 | 49.2 | 68.2 | 76.4 |
| Qwen2.5-VL$^+$-7B | 78.5 | 71.9 | 68.2 | 52.8 | 70.3 | 79.6 |
| Qwen2.5-VL$^+$-7B (●) | 40.3 | 62.7 | 61.8 | 45.3 | 65.0 | 75.2 |
| Qwen2.5-VL$^+$-7B (○) | 77.3 | 72.5 | 68.9 | 52.0 | 70.0 | 78.2 |

### 3.3.4 EFFECT OF GEOMETRIC

Building on SVE-Math-DeepSeek, we first train a projector while freezing LLM and visual encoder in the visual-language alignment stage using either AutoGeo, MAVIS, or GEOMETRIC image-caption alignment datasets. Training with GEOMETRIC yields +6.2% on perception tasks but suffer a substantial drop when training on other datasets ($\sim$-27% in Tab. 7). We then self-fine-tune the three pretrained models using MathV360K and instruction-formatted GEOMETRIC. Training with GEOMETRIC yields a notable +$\sim$4% on MathVerse and MathVista over SVE-Math-DeepSeek, whereas other variants underperform on MathVista.

To better assess the contribution of our perception-oriented GEOMETRIC, we conduct ablation studies by removing GEOMETRIC and using only mathematical visual reasoning datasets, such as MathV360K and MultiMath in the SFT stage. Additionally, we evaluate configurations where the visual encoder is frozen during the SFT stage, and only the projector and language model are updated. Tab. 8 shows that without GEOMETRIC, SFT on reasoning data alone yields only marginal improvements—or even performance degradation—compared to the base models. For example, Qwen2.5-VL$^+$-7B (●) performs 3.9% worse than Qwen2.5-VL-7B on MathVerse. This degradation may stem from overfitting to the reasoning dataset, resulting in excessive reliance on textual modality. In contrast, integrating GEOMETRIC into the training process enables the model to learn where to attend visually during reasoning, leading to improved overall performance. Moreover, comparisons between SVE-Math-DeepSeek$^+$ and SVE-Math-DeepSeek$^+$ (○), as well as Qwen2.5-VL$^+$-7B and Qwen2.5-VL$^+$-7B (○), further underscore the importance of optimizing the visual encoder to enhance visual perception.

## 4 CONCLUSION

Unlike semantically rich natural images, mathematical diagrams are structured and abstract, defined by symbolic elements and precise relationships. Current mathematical benchmarks often conflate reasoning and perception, making it difficult to assess true diagram understanding in MLLMs. To address this, we introduce MATHEMETRIC—a benchmark specifically designed for evaluating diagram perceptual capabilities. While these tasks appear trivial for humans to solve 'at a glance', they present significant challenges for current models. Through systematic evaluation across 20 MLLMs and detailed ablations, we identify several critical factors that degrade diagram perception, including text over-reliance, sensitivity to symbolic perturbations, and limited fine-grained spatial grounding. Our analyses further highlight the necessity of high-quality, structure-aware training data. Models trained with our GEOMETRIC dataset exhibit substantial improvements in perceptual accuracy and demonstrate measurable benefits on downstream reasoning benchmarks, bridging the gap between visual understanding and logical inference.

## 5 ETHICS STATEMENT

This work focuses on advancing diagram understanding and exploring cross-suite transfer to mathematical reasoning in multimodal large language models (MLLMs). Our datasets are either synthetically generated or derived from publicly available resources. For reused datasets, we adopt CLEVR (CC BY 4.0) and FigureQA (CC0 1.0); all reformatted files (*e.g.*, JSON conversions and lookup tables) are redistributed under the same licenses as their original sources. For GEOMETRIC, we will release the full package, including all generated data and the code used to produce it, under CC BY 4.0 upon acceptance, ensuring maximal reusability while preserving attribution to (a) the CLEVR and FigureQA authors and (b) our clause engine. This approach follows open-science best practices while safeguarding attribution and respecting licensing terms. Our work does not involve human subjects or sensitive information, and both models and datasets are intended solely to advance education and scientific discovery.

## 6 REPRODUCIBILITY STATEMENT

We have taken several steps to ensure the reproducibility of our work. Details of model architectures and evaluation protocols are provided in §2 and §3, with additional implementation details, training configurations, and hyperparameters included in Appendix §C. Our datasets are either publicly available (CLEVR, FigureQA) or synthetically generated; the data generation process and full reformatting details are described in Appendix §F. We release all code, data, evaluation benchmarks, and model checkpoints under a CC BY 4.0 license to maximize transparency and reusability.

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

# Math Blind: Failures in Diagram Understanding Undermine Reasoning in MLLMs

## Appendix

# A RELATED WORK

## A.1 MATHEMATICAL REASONING BENCHMARK

To evaluate MLLMs performance across different domains, various benchmarks Li et al. (2024); Yu et al. (2024); Li et al. (2023); Goyal et al. (2019); Liu et al. (2024) have been proposed, primarily focusing on natural scene understanding. However, benchmarks specifically designed for multimodal mathematical reasoning remain scarce.

Early benchmarks such as MathQA Amini et al. (2019), UniGeo Chen et al. (2022)Geometry3k Lu et al. (2021), GEOS Seo et al. (2015), and GeoQA++ Anand et al. (2024) introduced multimodal mathematical tasks but were limited in scope, often focusing on specific subdomains like plane geometry. More recent efforts have sought to provide broader and more diverse evaluations. MMMU Yue et al. (2024) assesses multimodal mathematical understanding with an emphasis on symbolic reasoning and word problem-solving. MathVista Lu et al. targets geometry-related tasks by integrating both real-world and synthetic diagrams to evaluate visual reasoning. MathVerse Zhang et al. (2024a) expands this by incorporating a wider range of multimodal challenges involving charts, graphs, and structured visual content. MATH-V Wang et al. (2025) further addresses the limitations of existing benchmarks by curating 3,040 high-quality math problems sourced from real-world competitions. While these benchmarks introduce visual elements, their core focus remains on assessing mathematical reasoning. It remains unclear whether the performance on these tasks truly reflects the models' ability to comprehensively understand the symbolic information in diagrams. The ability to accurately interpret mathematical symbols, diagrams, and spatial structures is a fundamental component of solving multimodal math problems, yet current benchmarks place limited emphasis on this aspect.

## A.2 MATHEMATICAL MLLMS

Recent multimodal large language models (MLLMs) Alayrac et al. (2022); Liu et al. (2023b); Sun et al. (2024) have made significant strides in vision-language understanding. However, foundation models such as GPT-4V Achiam et al. (2023b), Qwen2-VL Wang et al. (2024c), and Deepseek-VL2 Wu et al. (2024), while strong on general multimodal tasks, exhibit notable limitations in mathematical symbol recognition, spatial reasoning, and logical deduction—rendering them insufficient for vision-based mathematical problem solving.

Recent efforts have introduced math-specific MLLMs for improving mathematical reasoning. AlphaGeometry Trinh et al. (2024) achieves state-of-the-art results in geometry by leveraging theorem-proving and reinforcement learning. However, it relies solely on text-based diagram descriptions, lacking direct image processing. G-LLaVA Gao et al. (2023) extends LLaVA with geometric reasoning capabilities but struggles with complex visual structures and generalization beyond plane geometry. UniMath Liang et al. (2023) integrates structured math representations for solving visual word problems but remains focused on symbolic reasoning, limiting its handling of free-form mathematical diagrams. MatCha Liu et al. (2023a) specializes in chart-based reasoning, extracting quantitative relationships from structured visual data. However, its reliance on predefined formats limits adaptability to unstructured mathematical visuals.

Beyond architecture improvements, MAVIS Zhang et al. (2024b) introduces an automated data engine to generate large-scale mathematical visual datasets, reducing annotation costs while ensuring high-quality diagram-caption pairs and problem-solving rationales. However, its diagrams are constructed by simply combining basic geometric shapes, lacking mathematical relationship constraints such as perpendicularity and parallelism. AutoGeo Huang et al. (2024) considers special properties of lines, such as midlines and radii, as foundational to many geometric theorems, incorporating these properties into geometric figures. Reverse Chain-of-Thought (R-CoT) Deng et al. (2024) introduces

Table 9: Key statistics of MATHEMETRIC are summarized in Tab. 9a, and the subject-task distribution is illustrated in Fig. 9b.

| Statistic | Number |
|---|---|
| Total questions | 1,609 |
| - Multiple-choice questions | 893 (55.5%) |
| - Free-form questions | 406 (25.2%) |
| - True-false questions | 310 (19.3%) |
| Unique number of images | 1,198 |
| Unique number of questions | 1,514 |
| Unique number of answers | 380 |
| Total questions | 1609 |
| - Classification questions | 489 (30.4%) |
| - Counting questions | 401 (24.9%) |
| - Relation questions | 313 (19.5%) |
| - Grounding questions | 406 (25.2%) |
| Maximum question length | 200 |
| Maximum answer length | 4 |
| Maximum choice number | 4 |
| Average question length | 118.0 |
| Average answer length | 1.8 |
| Average choice number | 3.5 |
| Average question length | 16.09 |
| Average answer length | 1.21 |
| Average choice number | 3.40 |

(a)

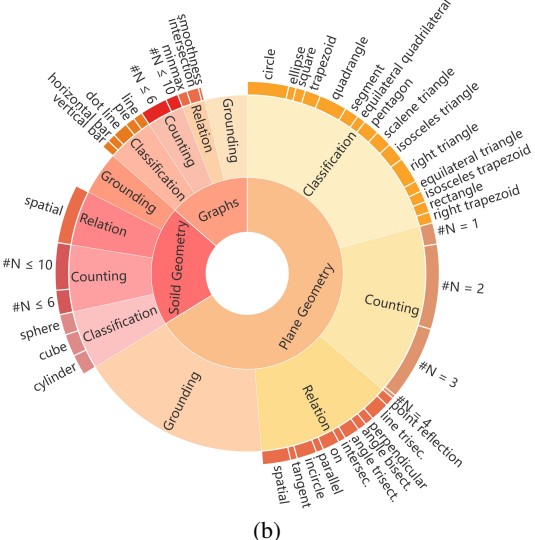

(b)

the Geometry Generation Chain to generate the geometry image and corresponding description. However, current data engines still fail to explore the underlying structures in mathematical diagrams, leading to ambiguous captions and redundant information that negatively impact MLLMs' perception abilities. Consequently, models trained on such datasets perform poorly on perception-demanding tasks, even falling below random guessing on multiple-choice questions. Visual Graph Arena Babaiee et al. (2025) corroborates this finding in the context of graph diagrams, where models fail to recognize that differently laid-out graphs represent the same underlying concept. These findings underscore the urgency of constructing structured symbolic visual datasets that enable models to both perceive and reason effectively.

## B    DETAILED DATA STATISTICS OF MATHEMETRIC

To comprehensively evaluate the perceptual abilities of MLLMs in mathematical contexts, MATHEMETRIC includes 1,609 questions grounded in 1,198 unique images. These span three key domains: plane geometry (66%), solid geometry (20%), and graphical representations (14%), such as line, bar, and pie charts (see Tab. 9). The benchmark comprises four core perception tasks: shape classification (489 questions, 30.4%), object counting (401, 24.9%), relationship identification (313, 19.5%), and fine-grained object grounding (406, 25.2%). Except for the grounding task, which uses free-form answers (*e.g.*, bounding box coordinates), all other tasks are framed as multiple-choice (55.5%) or true/false (19.3%) formats to support scalable and consistent evaluation. By default, ground truth answer is randomly assigned with equal probability, 25% for each choice letter and 50% for true/false. In our benchmark: the proportion of answer A/B/C/D is 26.7%/21.8%/28.3%/23.2%; and 48.3%/51.7% for true/false. As summarized in Tab. 9, the dataset includes 1,514 unique questions and 380 unique answers, with an average question length of 118.0 tokens and answer length of 1.8 tokens. This design supports fine-grained analysis of model performance across diverse perceptual challenges and subject domains.

## C    TRAINING IMPLEMENTATION DETAILS

We conduct experiments on both the math-specific SVE-Math-DeepSeek, and generic Qwen2.5-VL models (7B and 32B). Both follow a two-stage training pipeline: i) alignment training with image-caption pairs and ii) instruction tuning for reasoning and perception.

For SVE-Math-DeepSeek, we first train the vision-language projector using image-caption pairs (GEOMETRIC) while keeping the visual encoder and language model frozen. In the second stage, we perform full-parameter self-fine-tuning (SFT) using a combination of instructive GEOMETRIC

samples and MathV360K, updating the visual encoder, projector, and LLM jointly. For the generic Qwen2.5-VL-7B/32B models, we follow the same first-stage alignment process. To preserve prior reasoning capabilities and mitigate catastrophic forgetting, we adopt parameter-efficient LoRA tuning and incorporate an additional 300K MultiMath Peng et al. (2024) visual reasoning samples in the second SFT stage.

**Training Settings.** During alignment training, we use a global batch size of 256, a learning rate of $1e-4$, and train for one epoch with a maximum sequence length of 2048. In the SFT phase, we reduce the learning rate to $1e-5$ (with a vision-specific learning rate of $2e-6$) and maintain the same batch size and sequence length. We use the Adam optimizer without weight decay and apply a cosine learning rate schedule. To improve memory efficiency, we employ Fully Sharded Data Parallel (FSDP), gradient checkpointing, and enable BF16 precision. We avoid CPU/GPU offloading to maximize throughput.

**Hardware and Runtime.** All 7B model training was performed on $8 \times$ A100 GPUs (80GB each). Alignment training took approximately 5 hours, followed by 12 hours for full fine-tuning (SVE-Math-DeepSeek) and 16 hours for LoRA fine-tuning (Qwen2.5-VL-7B). For the 32B Qwen2.5-VL model, we used $16 \times$ H100 GPUs (96GB each), with alignment training taking 9 hours and LoRA-based SFT 15 hours.

**More Ablations.** 1) Herein, we assess whether it is necessary to count complex overlapping objects. Even humans interpret overlapping objects differently. To ensure more accurate conclusions for overlapping object counting, we asked three authors to manually and independently label the same 200 images sampled from our benchmark. The ground truth for these cases is represented as a list of one to three possible answers (if three annotators provided the same answer, we merged them). We added a system prompt to our counting templates: *you are required to count potential overlapping objects*. For evaluation, we adopt free-form answer matching—if the model's response matches any entry in the annotated answer list, it is considered correct. Top-1 accuracy is then computed based on this criterion, as shown in Tab 10.

Table 10: Accuracy (%) on non-overlapping *vs.* overlapping counting. Overlapping objects remain consistently harder for MLLMs.

| | LLaVA-v1.5-7B | LLaVA-v1.5-13B | G-LLaVA-7B | MultiMath-7B | Qwen2.5VL-7B | InternVL2.5-8B | InternVL2.5-38B | GPT-4o | SVE-Math-DeepSeek-7B | SVE-Math-DeepSeek+-7B |
|---|---|---|---|---|---|---|---|---|---|---|
| Non-overlapping | 43.8 | 44.5 | 45.8 | 35.9 | 54.8 | 40.0 | 56.2 | 57.1 | 40.1 | **88.2** |
| Overlapping | 28.7 | 29.1 | 29.3 | 18.4 | 32.8 | 21.7 | 31.9 | 38.9 | 23.7 | **67.3** |

2) Instead of using numeric-coordinate grounding, we adopt vertex lists as an alternative abstraction for grounding planar geometric objects. Vertex-list grounding is conceptually simpler than coordinate-based approaches: rather than predicting multiple continuous numerical values, the model only needs to identify discrete symbolic labels already annotated in the diagram. For example, *Q:* Please provide the list of vertex labels described by the sentence: Square. *A:* ABCD. A comparison between vertex-list grounding and numeric-coordinate grounding is shown in Tab. 11.

Table 11: Comparison of vertex-list grounding *vs.* numeric-coordinate grounding for planar diagrams.

| | Qwen2-VL-7B | Qwen2.5-VL-7B | LLaVA-v1.5-7B | InternVL2.5-32B | GPT-4o | G-LLaVA-7B | MultiMath-7B | SVE-Math-DeepSeek-7B | SVE-Math-DeepSeek+-7B |
|---|---|---|---|---|---|---|---|---|---|
| Vertex | 45.6 | 49.1 | 1.8 | 42.7 | 44.8 | 3.9 | 33.1 | 41.7 | **89.3** |
| Numerics | 12.8 | 18.5 | 14.2 | 2.5 | 1.1 | 0.4 | 1.1 | 3.6 | **82.9** |

3) To prevent models from exploiting vertex-count shortcuts in the shape-classification task, we design hard distractors for 75% of the questions. For each ground-truth polygon, at least one distractor is drawn from the same polygon family, ensuring that models cannot rely solely on the number of vertices. For instance, when the correct answer is a scalene triangle, distractors are sampled from other triangle types (isosceles, right, equilateral). Likewise, for quadrilaterals, distractors include shapes such as rectangles, squares, or trapezoids (*e.g.*, a rectangle paired with a right trapezoid; see Fig. 9, first row). To further validate that vertex-count shortcuts are not driving model performance, we construct an additional controlled dataset of 200 planar diagrams in which vertex labels are replaced with purely visual numeric markers (*e.g.*, 1, 2, 3 $\cdots$). These markers do not encode polygon

Figure 8: Visualization of samples where objects are annotated using numerical visual markers instead of vertex labels. Zoom in for the best view.

type and therefore eliminate vertex-count cues. Example cases are provided in Fig. 8, and the corresponding ablation results are summarized in Tab 12. These results show diverse behavior across models when switching from vertex labels to visual markers. If vertex-count shortcuts dominated model performance, all models would exhibit similar, substantial drops under the marker condition—but this is not observed. Several models (*e.g.*, Qwen2.5-VL-7B, LLaVA-v1.5-7B, SVE-Math-DeepSeek-7B, and our model) maintain stable performance under both conditions. The observed irregularities instead reflect differences in models' ability to interpret visual markers: for example, GPT-4o handles marker-based diagrams well, whereas InternVL2.5-38B performs better with vertex labels. These findings confirm that our distractor design is robust and that model performance is not driven by vertex-count shortcuts.

Table 12: Accuracy (%) on vertex-list *vs.* visual-marker classification for planar diagrams.

| | Qwen2-VL-7B | Qwen2.5-VL-7B | LLaVA-v1.5-7B | InternVL2.5-38B | GPT-4o | G-LLaVA-7B | MultiMath-7B | SVE-Math-DeepSeek-7B | SVE-Math-DeepSeek$^{+}$-7B |
|---|---|---|---|---|---|---|---|---|---|
| Vertex | 50.7 | 48.6 | 34.6 | 55.3 | 53.1 | 31.6 | 44.1 | 43.9 | 72.6 |
| Markers | 43.3 | 47.9 | 35.3 | 43.3 | 54.8 | 28.8 | 33.2 | 41.1 | 71.7 |

**Discussion of MINT-CoT and Vision-R1 *vs.* Ours.** MINT-CoT and Vision-R1 align with our findings that perception is critical for reasoning, yet operate at different scopes:

1) Relation to MINT-CoT: MINT-CoT explicitly injects interleaved visual tokens into the chain-of-thought, enabling the model to reference visual evidence during reasoning. This represents an important direction for coupling perception and reasoning step-wisely. In contrast, our work focuses on improving diagram perception itself, without introducing any specialized mechanism for mixing visual and textual tokens in the reasoning process. As a result, the two lines of work are compatible: MINT-CoT studies how visual information is integrated into CoT reasoning; our work studies whether the model can accurately perceive symbolic diagrams in the first place, and how enhanced perception naturally transfers to reasoning.

2) Relation to Vision-R1: Vision-R1 aims to incentivize reasoning ability on top of a strong base model using reinforcement learning. A key factor in Vision-R1's success is the quality and diversity of its cold-start data: over 43 curated datasets spanning mathematical diagrams, science and medical figures, general QA images, and figure-understanding datasets. Importantly, during data construction, Vision-R1 uses image captions as CoTs, giving the model holistic visual context that supports the development of high-quality reasoning traces. While both works support the idea that perception is a prerequisite for effective multimodal reasoning, Vision-R1 focuses primarily on improving reasoning ability, whereas our work focuses on explicitly enhancing symbolic diagram perception. **This distinction is crucial: reasoning improvements in Vision-R1 rely heavily on RL strategies and large-scale, high-quality CoT training samples, not on explicit perceptual enhancement.**

Overall, we summarize the key points below:

- Our work: focuses on improving diagram perception and demonstrates measurable transfer from enhanced perception to downstream reasoning.

- MINT-CoT: interleaves visual tokens within chain-of-thought reasoning to produce more visually aligned inference.

- Vision-R1: incentivizes reasoning ability through reinforcement learning, built on high-quality, large-scale perception–reasoning training data.

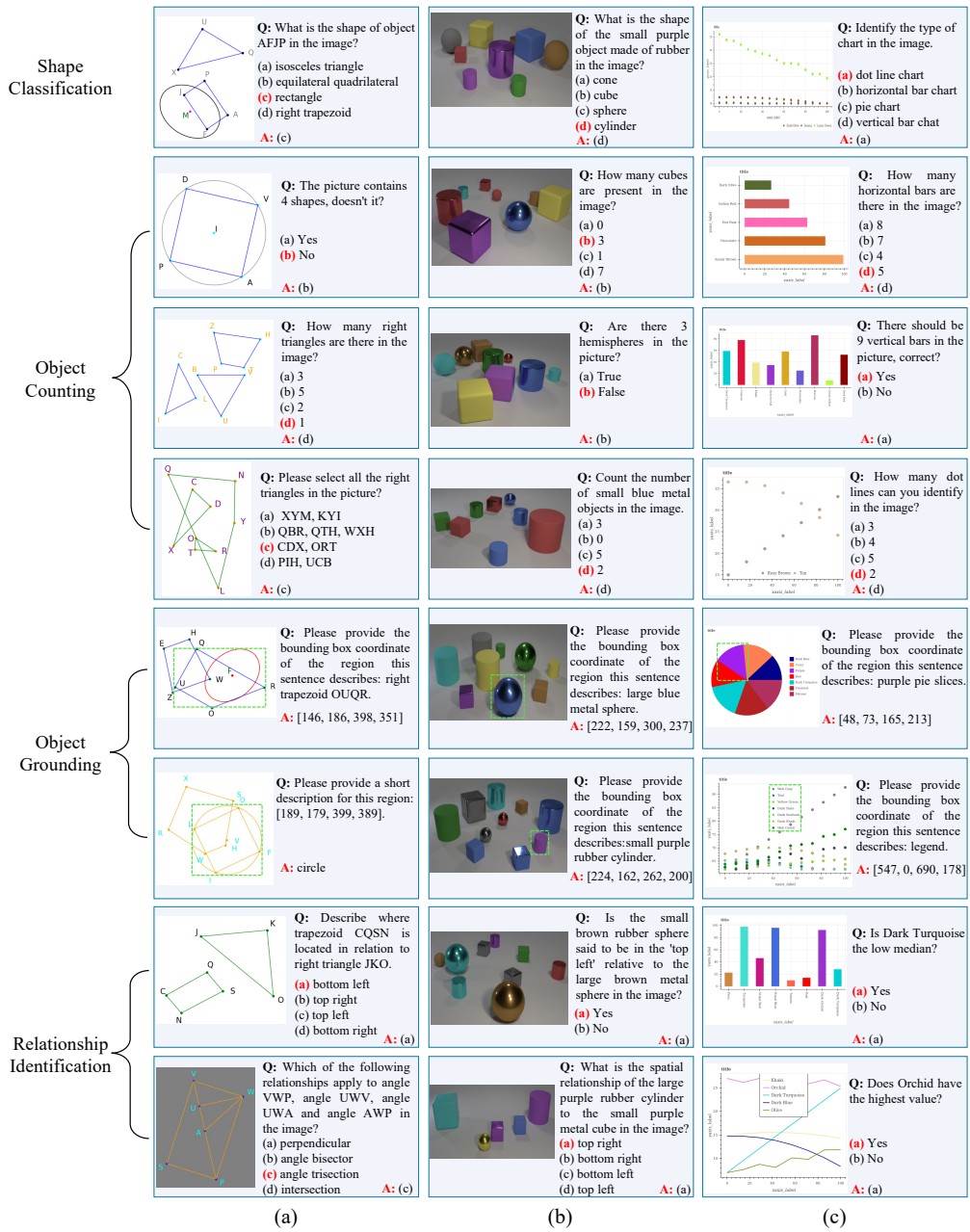

Figure 9: Visualization of sample cases from MATHEMETRIC. (a), (b), and (c) correspond to problems related to Plane Geometry, Solid Geometry, and Graphs, respectively.

## D    VISUALIZATION OF MATHEMETRIC

MATHEMETRIC is specifically designed to evaluate perception-demanding tasks in Multimodal Large Language Models (MLLMs), focusing on four core visual understanding capabilities: shape classification, object counting, object grounding, and relationship identification. These tasks, commonly studied in classical computer vision, are typically solvable by humans with minimal cognitive effort. Our benchmark spans three key domains—plane geometry, solid geometry, and mathematical graphs—to ensure broad coverage of visual representations encountered in educational and scientific contexts. Fig. 9 provides illustrative examples from MATHEMETRIC, demonstrating the diagrammatic input and corresponding question–answer (Q&A) pairs used for evaluation.

## E PROBLEM TEMPLATES

This section introduces the problem templates employed in MATHEMETRIC, with illustrative examples provided in Tab. 13. We adopt a template-based generation engine to systematically construct diverse perception-oriented Q&A pairs. Each question is generated by parsing the structured JSON annotations of a given image, which include information on shapes, object attributes, spatial positions, and inter-object relationships. These elements are then filled into carefully designed templates (Tab. 13) to produce grammatically correct and semantically meaningful Q&A pairs.

Our template designs are tailored to accommodate diverse subject domains—including plane geometry, solid geometry, and graph-based diagrams—and span a spectrum of perceptual complexity, from coarse-level to fine-grained tasks. For example, a coarse-level shape classification template may pose a question such as, "What is the shape of the object with {vertices}?", where the correct answer is directly retrieved from the structured annotations. For fine-grained object localization, we adopt a grounding template similar to that used in Wu et al. (2024), prompting models with: "Please provide the bounding box coordinates of the region this sentence describes: {shape} {vertices}". This allows us to evaluate the model's ability to extract precise spatial information and align language with visual primitives at a granular level.

We design three types of questions: multiple-choice, true/false, and open-ended. Specifclly, for multiple-choice questions, we incorporate plausible distractors to challenge the model's geometric perception. These distractors are selected from geometric candidate pools, varying from visually similar to clearly distinguishable options, ensuring a balanced evaluation of fine-grained visual discrimination. By leveraging this template engine, MATHEMETRIC ensures consistent question formulation, controlled variation in difficulty, and robust coverage of geometric primitives and their relationships, forming a reliable testbed for evaluating diagram perception in MLLMs.

## F DATASET CONSTRUCTION FOR PLANE/SOLID GEOMETRY & GRAPHS

### F.0.1 SYNTHETIC DATA ENGINE FOR PLANE GEOMETRY

Fig. 4 (main paper) describes the entire generation process. Inspired by AlphaGeometry Trinh et al. (2024), we use geometric clauses as fundamental units to construct complex plan geometric figures. A geometric clause is a formalized description of basic geometric objects and mathematical relationships, along with their properties or attibutes, *a.k.a.* prerequisite points. We first construct two geometry substrate pools, one containing 16 different geometric shapes (*i.e.*, isosceles triangle, square, rectangle, parallelogram, isosceles trapezoid, pentagon, circle ... ellipse and segment), and the other defining 10 mathematical relationships (*i.e.*, on, intersection, parallel, perpendicular, tangent, ... and reflection). We then randomly sample one or more substrates from these pools and pass them through a verifier, which ensures that logically paired shapes and relationships are preserved in the construction of valid geometric images. The verifier makes decisions based on either manually designed rules, fundamental mathematical knowledge, or prerequisite points. For example, parallel lines cannot intersect, and without enough prerequisite points, an angle trisection clause is invalid. In general, the chosen relationships are enforced by introducing additional shapes into the figure, ensuring that the specified relationships are accurately maintained and geometrically consistent throughout the construction process. We finally save the outputs as structured JSON annotations for image rendering using the Matplotlib package Hunter (2007) and generating question-answer pairs via template-based pipelines.

**Image Generation.** The JSON annotations define foreground and background styles, *i.e.*, colors sampled from a monochromatic palette, line width, and font size, as well as object shape information (class names), attributes (vertices labeled with random letters), bounding box locations, and mathematical relationships. Notably, spatial relationships are generated based on the bounding box locations of two objects (*e.g.*, top-left, top-right, bottom-left, bottom-right). Similar to the image rendering process in Trinh et al. (2024), we translate geometric clauses into visual representations using Python code. This process first determines coordinates of points defined in each clause for basic shape visualization, then generates new primitives based on specified relationships. To increase task difficulty, we apply image augmentations, *e.g.*, Gaussian noise, irregular scribbles, and wedge-shaped symbols for congruent angles or auxiliary lines (see Figs. 10 and 12).

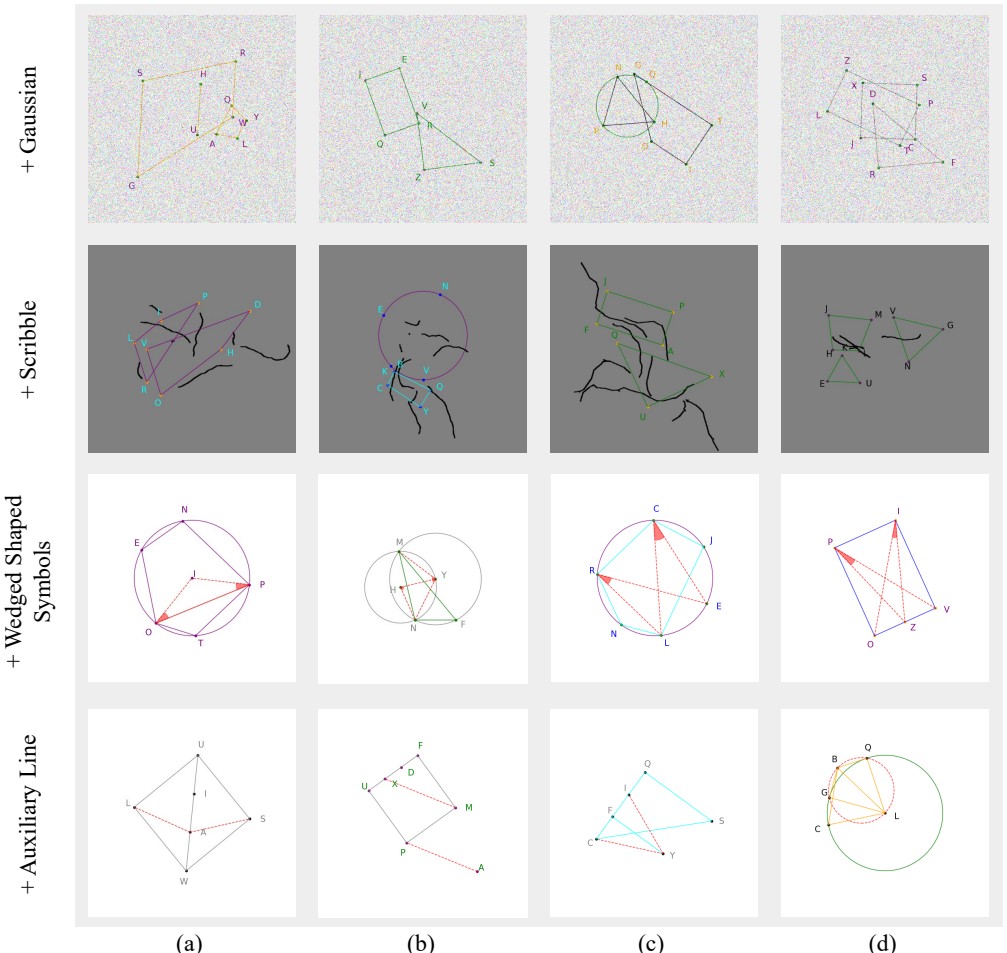

Figure 10: Visualizing distractors *w.r.t.* adding Gaussian noise, drawing irregular scribbles and incorporating wedge-shaped symbols or auxiliary lines.

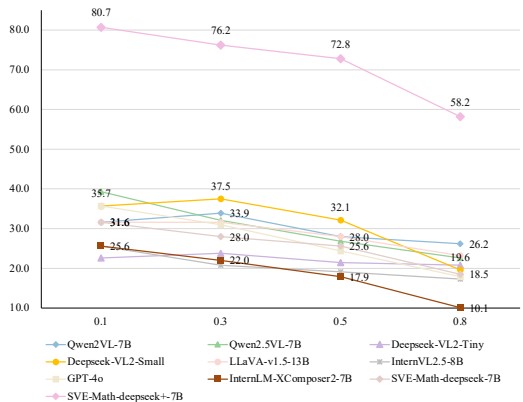

Figure 11: Perceptual perfromance *w.r.t.* varying Gaussian noise variance.

**Question-Answer Generation.** Based on structured annotations, we employ a template-based pipeline to generate multiple-choice and true-false questions across four perception tasks. For example, in constructing Q&A pairs for shape classification, we first randomly select an object and its associated attributes from the given figure. We then formulate a question by filling placeholders

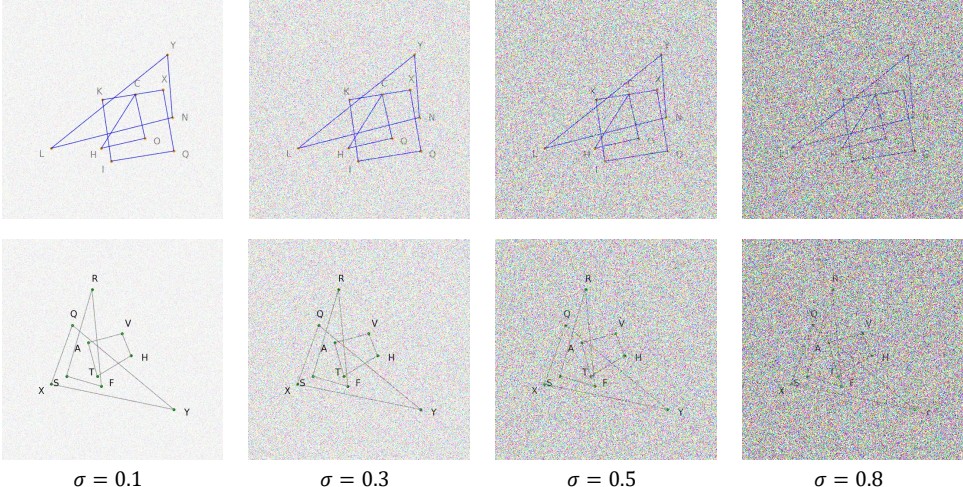

$\sigma = 0.1$        $\sigma = 0.3$        $\sigma = 0.5$        $\sigma = 0.8$

Figure 12: Visualization of distorted images under varying levels of Gaussian noise.

in our carefully designed templates. For true-false questions, the ground truth answer is directly derived from the JSON annotations. For multiple-choice questions, we generate plausible distractors to challenge the model's geometric perception, combining them with the correct answer into a set of four choices. Distractors are selected from geometric candidate pools, ranging from visually similar (*e.g.*, equilateral *vs.* isosceles triangles) to visually dissimilar pairs (*e.g.*, equilateral triangle *vs.* circle). Further details on the Q&A generation templates are provided in Tab. 13.

Copyright and Consent: For GEOMETRIC, we will release the entire package including all data and code used to generate samples under CC BY 4.0 on acceptance.

### F.0.2 REFORMAT DATASET FOR SOLID GEOMETRY & GRAPHS

Copyright and Consent: CLEVR Johnson et al. (2017) is under CC BY 4.0 and FigureQA Kahou et al. (2017) is under CC0 1.0. All files that we re-publish (JSON reformats and look-up tables) will carry the same licenses as their sources.

**Solid Geometry.** CLEVR Johnson et al. (2017) is a synthetic Visual Question Answering (VQA) dataset containing 3D-rendered objects. Each object in the scene is defined by its 3D position and a set of attributes, including size (small or large), shape (cube, cylinder, sphere), material (rubber or metal), and color (gray, blue, brown, yellow, red, green, purple, cyan). However, CLEVR's annotation format is incompatible with our template-based pipeline for generating perception evaluation QA pairs. To resolve this, we reformat the original annotations into structured JSON, following the same format as the synthetic data process. Specifically, we extract object information from CLEVR's scene annotations and organize them into shape information and object attributes. We then leverage an object's 3D location and its direction for calculating its 2D top-left and bottom-right coordinates, from which relative spatial relationships are derived.

Once the reformatted JSON annotations are generated, we apply the same template-based QA generation process as designed for plane geometry, with an additional uniqueness check for shape classification. To ensure clear and consistent labeling, we filter out ambiguous cases where identical attribute combinations correspond to different shapes within the same image. For example, if an image contains multiple distinct shapes with the same attributes (*e.g.*, a large rubber blue square and a large rubber blue cylinder), we exclude classification questions that rely on the attribute combination 'large rubber blue' to prevent ambiguity.

**Graphs.** FigureQA Kahou et al. (2017) is a visual reasoning dataset containing over one million question-answer pairs, grounded in synthetic, scientific-style figures, including line plots, dot-line plots, vertical and horizontal bar graphs, and pie charts. The official annotations capture various relationships between plot elements and evaluate characteristics such as maximum, minimum, smooth-

ness, and intersection, all framed as binary yes/no questions. Additionally, FigureQA provides numerical data annotations used to generate each figure, along with bounding-box annotations for all plot elements. Similar to CLEVR, we reformat FigureQA annotations compatible with template-based pipelines.

Specifically, the shapes include five graph types as defined in the official specification, with each element's attribute represented by its unique color. Additionally, we store the bounding box coordinates of each foreground element (such as lines, bars, and pie slices), along with legends and titles, as ground truth for the grounding task. For relationships, we follow the binary true/false question-answering problems to evaluate the model's understanding of geometric and graphical relationships.

## G   EXAMPLE ILLUSTRATION OF GEOMETRIC

Figs. 13-15 demonstrate how GEOMETRIC delivers explicit geometric information—covering object count, shape classification, fine-grained bounding box coordinates, and inter-object relationships—presented in both caption-style and instruction-following conversational formats.

## H   VISUAL DISTRACTORS

To evaluate the robustness of MLLMs' perceptual capabilities, we introduce a set of visual perturbations through data augmentation. These include Gaussian noise, irregular scribbles, wedge-shaped symbols, and auxiliary lines, as shown in Fig. 10. The aim is to simulate real-world visual ambiguities and assess whether MLLMs can retain accurate geometric understanding under degraded conditions. These controlled distortions provide a rigorous benchmark for evaluating the models' ability to extract and interpret mathematical structures from visually complex inputs.

By gradually increasing the Gaussian noise level, we can systematically evaluate its impact on the ability of MLLM to recognize mathematical structures. As shown in Fig. 12, the image transitions from a noise level of 0.1 to 0.8, progressively blurring the geometric features. We observe that as the noise intensity increases, the difficulty of recognizing mathematical structures also rises, posing greater challenges to accurate recognition and reasoning. When the noise level reaches $\sim \mathcal{N}(\mathbf{0}, \mathbf{0.8})$, the mathematical structures become nearly indistinguishable to the human eye, resulting in a significant performance drop on MATHEMETRIC, as illustrated in Fig. 11.

## I   CASE STUDY

**Model Responses.** In Figs. 16-18, we present a comparative analysis of model responses across several Multimodal Large Language Models (MLLMs), including SVE-Math-DeepSeek Zhang et al. (2025), InternVL2.5 Chen et al. (2024a), Qwen2.5-VL Bai et al. (2025), and GPT-4o, alongside our enhanced variants: SVE-Math-DeepSeek$^+$ and Qwen2.5-VL$^+$. Our evaluation shows that fine-tuning base models (SVE-Math-DeepSeek and Qwen2.5-VL) on GEOMETRIC significantly improves response accuracy over their respective baselines. This improvement suggests that our carefully curated training data effectively enhances the model's mathematical perception, enabling it to better comprehend problem structures, reason through mathematical concepts, and generate more precise answers. For example, in Fig 16 (a), the base model fails to recognize the hypotenuse, while the GEOMETRIC model correctly identifies it and naturally applies the Pythagorean theorem to reach the correct answer. Similarly, in Fig 16 (b), the GEOMETRIC model correctly perceives the diameter, infers the presence of a right triangle, and again applies the appropriate theorem. In Fig 18 (a), both the base model and ours know the relevant theorem—the Inscribed Angle Theorem. However, the base model misperceives key visual cues (*e.g.*, tangent lines), leading to hallucinated angle relations and an incorrect final answer. The GEOMETRIC model correctly perceives the tangency and the right-angle structure, enabling it to execute the theorem correctly and arrive at the correct answer. Furthermore, we observe that even GPT-4o, a strong competitor in the field, produces incorrect responses in certain cases. Upon closer examination, these errors often stem from inaccurate mathematical perception rather than purely computational mistakes. This observation reinforces the notion that an MLLM's ability to accurately perceive and interpret mathematical content is a critical factor in achieving high performance. Overall, our findings highlight the fundamental role of precise

perception in mathematical reasoning, akin to its importance in vision and language understanding. perception can lead to substantial gains in accuracy and reliability.

**Error Examples.** In this section, we provide more detailed error examples of GPT-4o, Qwen2.5-VL-7B, and our 7B model. We categorize the errors into two types: Chain-of-Thought (CoT) Errors and Recognition Errors. CoT errors occur when the model engages in step-by-step reasoning for perception questions but ultimately provides incorrect answers. Recognition errors, on the other hand, arise when the model attempts direct answering without reasoning yet fails to produce the correct result. Representative examples for each error type are illustrated in Figs. 19-28.

## J    LIMITATIONS AND BROADER IMPACTS

**Limitations.** The proposed MATHEMETRIC focuses on evaluating MLLMs' perceptual capabilities in structured, symbolic mathematical diagrams spanning plane geometry, solid geometry, and graphical representations. It enables fine-grained assessment through carefully designed perception tasks. Our findings clearly show that current models exhibit limited perceptual ability in diagram understanding across all three subjects, but our proposed training dataset, GEOMETRIC, remains limited to synthetic and geometry-based content. As such, it may not fully capture the diversity and ambiguity of real-world educational materials (*e.g.*, textbook figures, handwritten notes, or scanned diagrams). In future work, we aim to extend our graph-based data construction to solid and graphical domains, which requires further investigation into their distinct structural representations. For instance, in graphical images such as line and bar charts, the underlying structure differs significantly from geometric diagrams. Nodes may correspond to data points, axis labels, or bars, while edges may capture trends, groupings, or relational mappings across axes. Developing meaningful node–edge representations in these contexts will be essential for enabling accurate visual grounding and interpretation. Moreover, we believe additional tasks, such as visual marker detection, are also critical for comprehensively evaluating MLLMs. For instance, recognizing special diagrammatic markers that represent relationships like parallelism or perpendicularity (*e.g.*, double lines or small squares) is essential for deeper geometric understanding. Similarly, OCR capabilities are important, such as detecting angle measures or textual annotations embedded within diagrams. Expanding the benchmark to cover these aspects would further advance the assessment of MLLMs' diagram perception abilities.

Furthermore, although we analyze how perception affects reasoning, our current setup does not explicitly model the interleaved interaction between visual grounding and logical reasoning—a critical direction for advancing multimodal mathematical reasoning.

**Broader Impacts.**    This paper sheds light on a critical yet underexplored capability of MLLMs—mathematical diagram perception—and aims to push the field toward more interpretable and reliable multimodal reasoning. By isolating perception from reasoning, we reveal fundamental limitations in current models and provide a diagnostic tool for future model development. These insights could benefit applications in math education, AI-assisted learning, and cognitive science research. Caution is warranted, however, as over-reliance on synthetic datasets may lead to overfitting, and enhanced perceptual accuracy does not inherently ensure reliable reasoning. Furthermore, improved performance on benchmark tasks may not directly translate to robust generalization in real-world scenarios. We advocate for future research that bridges synthetic benchmarks with naturalistic data and advances human-aligned evaluation frameworks for visual reasoning systems.

## K    USE OF LARGE LANGUAGE MODELS (LLMS)

We used LLMs solely as an auxiliary tool for grammar correction and minor language polishing. They did not contribute to research ideation, experimental design, data analysis, or writing of technical content.

| *Structured Descriptions* | *Multi-turn Conversation* |
|---|---|
| **Image:** 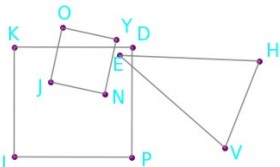 | **Image:** 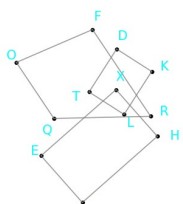 |
| **Question:** Please provide structured geometric information for the image. **Answer:** First, I count 3 prominent objects in the image, including 1 rectangle, 1 isosceles triangle, 1 square. Next, for the shape information, object DPIK is a rectangle, object EHV is an isosceles triangle, object JOYN is a square. Furthermore, I also know the fine-grained bounding box coordinates: the rectangle DPIK is located at [113, 205, 251, 328], the isosceles triangle EHV is located at [236, 215, 399, 317], the square JOYN is located at [154, 179, 230, 256]. Finally, let me explain the relationships: rectangle DPIK is in the bottom left relative to isosceles triangle EHV; rectangle DPIK is in the bottom left relative to square JOYN. | **Question:** Provide the shape type of object DKLT in this image. **Answer:** rectangle  **Question:** Count the number of rectangles in the image. **Answer:** 2  **Question:** Please provide the bounding box coordinate of the region this sentence describes: rectangle EXHI. **Answer:** [261, 143, 364, 251]  **Question:** Describe the relationship of rectangle EIHX relative to trapezoid FRQO in the image. **Answer:** bottom right |

Figure 13: Sampels of caption-style and instruction-following GEOMETRIC.

| *Structured Descriptions* | *Multi-turn Conversation* |
|---|---|
| **Image:** 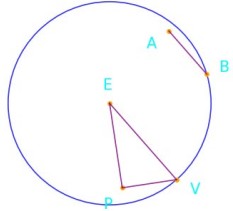 | **Image:** 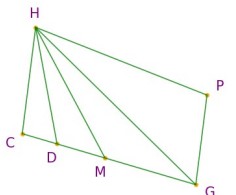 |
| **Question:** Please provide structured geometric information for the image.  **Answer:** First, I count 2 prominent objects in the image, including 1 right triangle, 1 circle. Next, for the shape information, object EVP is a right triangle, object E is a circle. Furthermore, I also know the fine-grained bounding box coordinates: the right triangle EVP is located at [256, 256, 348, 374], the circle E is located at [113, 113, 399, 399]. Finally, let me explain the relationships: right triangle EVP is in the bottom right relative to circle E; line AB is parallel to line EV. | **Question:** Can you identify the type of object CHPG in the picture? A:equilateral quadrilateral B:trapezoid C:right trapezoid D:right triangle  **Answer:** B  **Question:** How many trapezoids are there in the image? A:3 B:4 C:1 D: 2  **Answer:** C  **Question:** Please provide the bounding box coordinate of the region this sentence describes: trapezoid CHPG.  **Answer:** [113, 133, 399, 380]  **Question:** Does the image show the relationship among angle CHG, angle CHD, angle DHM and angle MHG as "angle trisection"? A:Yes B:No.  **Answer:** A |

Figure 14: Sampels of caption-style and instruction-following GEOMETRIC.

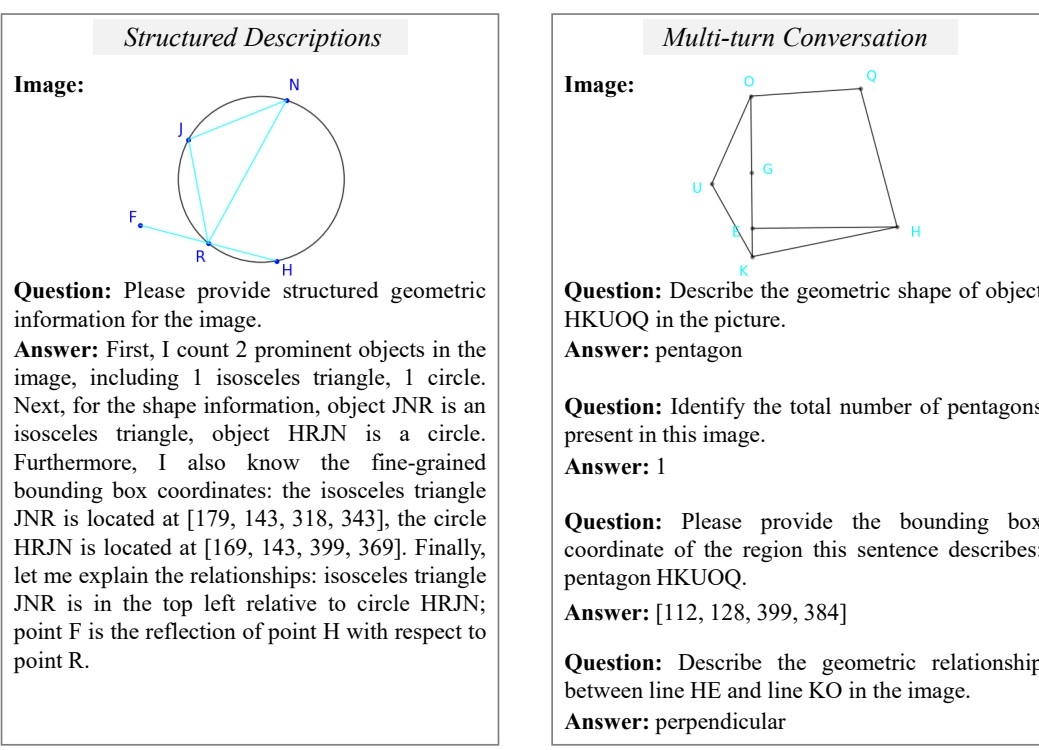

Figure 15: Sampels of caption-style and instruction-following GEOMETRIC.

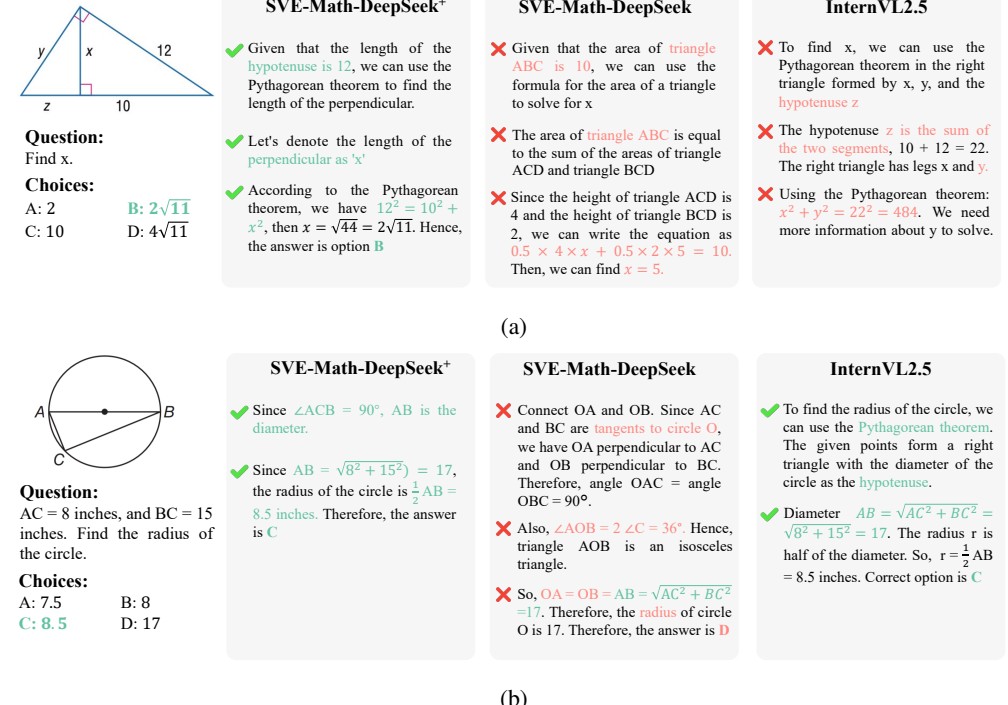

Figure 16: Response comparisons between SVE-Math-DeepSeek$^+$ (7B), SVE-Math-DeepSeek (7B), and InterVL2.5 (8B) in MathVerse.

Table 13: Examples of problem templates used by MATHEMETRIC on different source data acorss different tasks.

| Source | Task | Three randomly chosen examples from hundreds. |
|---|---|---|
| Plane Geometry | cls | What is the shape of object {vertices} in the image? Choices: A:{a} B:{b} C:{c} D:{d}.
Can you identify the type of object {vertices} in the picture? Choices: A:{a} B:{b} C:{c} D:{d}.
Can you identify the type of object in the picture? Choices: A:{a} B:{b} C:{c} D:{d}.
What is the shape of the object in the image? Choices: A:{a} B:{b} C:{c} D:{d}. |
| | cnt | There {be} only {num} {shape} in the picture, right? Choices: A:Yes B:No.
{be} there only {num} {shape} in the picture? Choices: A:Yes B:No.
You can only see {num} objects in the picture, can't you? Choices: A:Yes B:No.
There should be only {num} shapes in the picture, correct? Choices: A:Yes B:No.
How many {shape}s can you find in the picture? Choices: A:{a} B:{b} C:{c} D:{d}.
Please count all the {shape}s in the image. Choices: A:{a} B:{b} C:{c} D:{d}.
What is the total number of shapes in the picture? Choices: A:{a} B:{b} C:{c} D:{d}.
How many objects can you identify in the image? Choices: A:{a} B:{b} C:{c} D:{d}.
Please identify and select all the {shape}s in the picture. Choices: A:{a} B:{b} C:{c} D:{d}.
Find and select all the {shape}s in the picture. Choices: A:{a} B:{b} C:{c} D:{d}. |
| | grd | Please provide the bounding box coordinate of the region this sentence describes: {shape} {vertices}. |
| | rlat | Can the relationship {preposition} {shape} in the image be described as "{relation}"? Choices: A:Yes B:No.
Does the image show the relationship {preposition} {shape} as "{relation}"? Choices: A:Yes B:No.
Is {shape1} described as being in the '{relation}' relative to {shape2} in the image? Choices: A:Yes B:No.
Is {shape1} said to be in the '{relation}' relative to {shape2} in the image? Choices: A:Yes B:No.
What is the relationship {preposition} {shape} in the picture? Choices: A:{a} B:{b} C:{c} D:{d}.
Can you identify the relationship {preposition} {shape} in the image? Choices: A:{a} B:{b} C:{c} D:{d}.
What is the relative position of {shape1} to {shape2} in the image? Choices: A:{a} B:{b} C:{c} D:{d}.
What is the spatial relationship of shape1 to shape2 in the image? Choices: A:{a} B:{b} C:{c} D:{d}. |
| Soild Geometry | cls | What is the shape of the {size} {color} object made of {material} in the image? Choices: A:{a} B:{b} C:{c} D:{d}.
Can you identify the type of {size} {color} object with {material} material in the picture? Choices: A:{a} B:{b} C:{c} D:{d}. |
| | cnt | There {be} only {num} {shape} in the picture, right? Choices: A:Yes B:No.
{be} there only {num} {shape} in the picture? Choices: A:Yes B:No.
There are only {num} objects in the picture, right? Choices: A:Yes B:No.
Are there only {num} shapes in the picture? Choices: A:Yes B:No.
How many {shape}s can you find in the picture? Choices: A:{a} B:{b} C:{c} D:{d}.
Please count all the {shape}s in the image. Choices: A:{a} B:{b} C:{c} D:{d}.
Count the shapes in the image. Choices: A:{a} B:{b} C:{c} D:{d}.
How many shapes can you visually identify in the image? Choices: A:{a} B:{b} C:{c} D:{d}.
How many {size} {color} {material} objects are there in the image? Choices: A:{a} B:{b} C:{c} D:{d}.
How many {size} {color} {material} objects are present in the image? Choices: A:{a} B:{b} C:{c} D:{d}. |
| | grd | Please provide the bounding box coordinate of the region this sentence describes: {size} {color} {material} {shape}. |
| | rlat | Is it correct that {shape1} is described as being in the '{relation}' relative to {shape2} in the image? Choices: A:Yes B:No.
Is {shape1} described as being in the '{relation}' relative to {shape2} in the image? Choices: A:Yes B:No.
Is {shape1} said to be in the '{relation}' relative to {shape2} in the image? Choices: A:Yes B:No.
Can you confirm that {shape1} is in the '{relation}' relative to {shape2} in the image? Choices: A:Yes B:No.
In the image, where is {shape1} in relation to {shape2}? Choices: A:{a} B:{b} C:{c} D:{d}.
What is the relative position of {shape1} to {shape2} in the image? Choices: A:{a} B:{b} C:{c} D:{d}.
What is the spatial relationship of {shape1} to {shape2} in the image? Choices: A:{a} B:{b} C:{c} D:{d}.
Describe how {shape1} is situated relative to {shape2} in the image. Choices: A:{a} B:{b} C:{c} D:{d}. |
| Graphs | cls | What type of chart is shown in the image? Choices: A:{a} B:{b} C:{c} D:{d}.
Identify the type of chart in the image. Choices: A:{a} B:{b} C:{c} D:{d}.
Which of the following best describes the chart graph in the image? Choices: A:{a} B:{b} C:{c} D:{d}.
Which kind of chart does the image represent? Choices: A:{a} B:{b} C:{c} D:{d}.
What kind of data visualization is shown in the image? Choices: A:{a} B:{b} C:{c} D:{d}.
Can you identify the type of chart in the image? Choices: A:{a} B:{b} C:{c} D:{d}. |
| | cnt | {be} there only {num} {shape} in the picture? Choices: A:Yes B:No.
The picture contains only {num} {shape}, each represented by a different color, doesn't it? Choices: A:Yes B:No.
You can only see {num} {shape} with different colors in the picture, can't you? Choices: A:Yes B:No.
How many {shape}s are there in the image? Choices: A:{a} B:{b} C:{c} D:{d}.
Can you count the total number of {shape}s in the image? Choices: A:{a} B:{b} C:{c} D:{d}.
What is the total number of {shape}s in the picture? Choices: A:{a} B:{b} C:{c} D:{d}. |
| | grd | Please provide the bounding box coordinate of the region this sentence describes: {color} {shape}. |
| | rlat | {question} Choices: A:Yes B:No. |

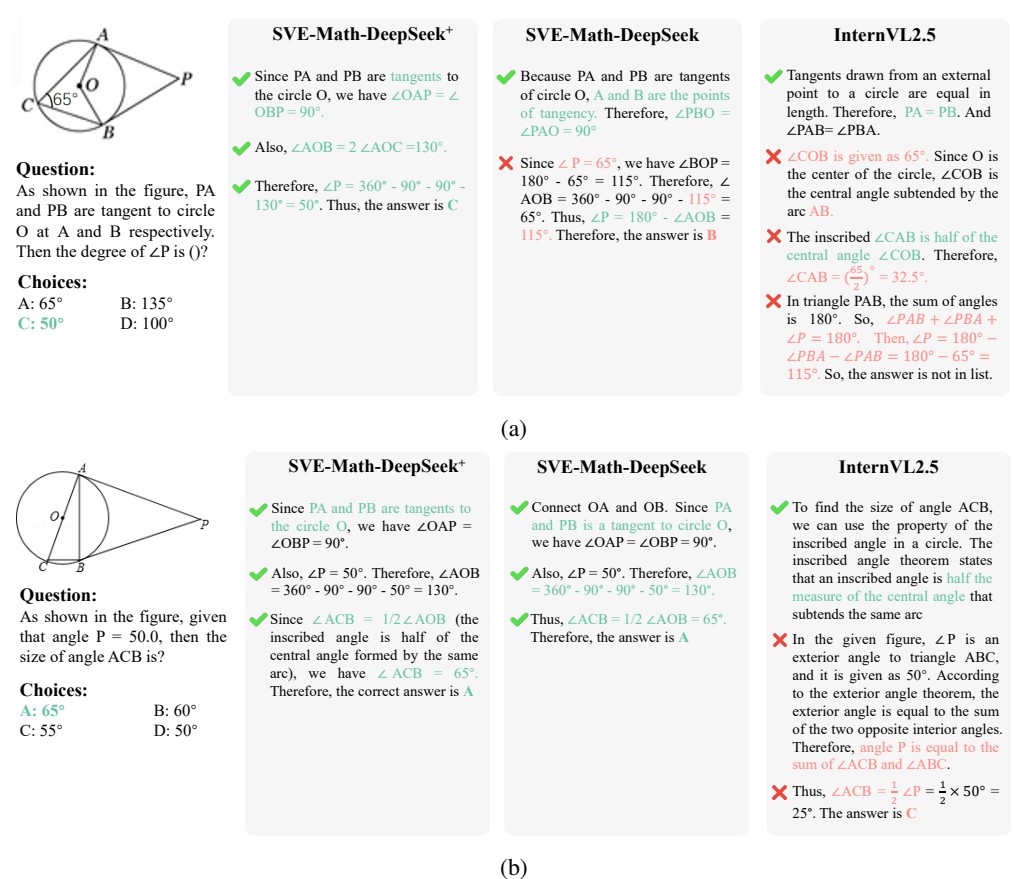

Figure 17: Response comparisons between SVE-Math-DeepSeek$^+$ (7B), SVE-Math-DeepSeek (7B), and InterVL2.5 (8B) in MathVerse.

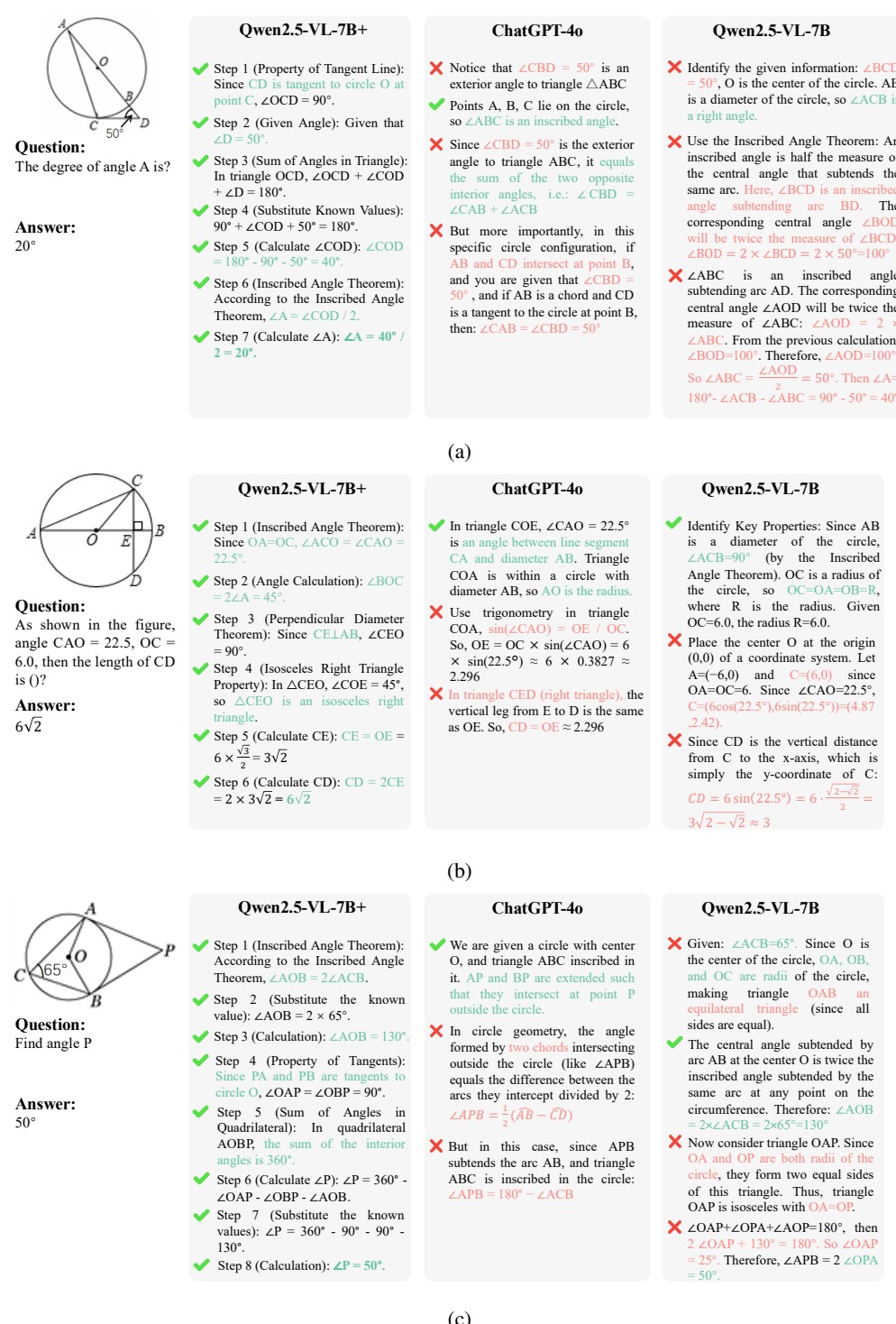

Figure 18: Response comparisons between Qwen2.5-VL-7B+ (Qwen2.5-VL-7B fine-tuned with GEOMETRIC using LoRA), GPT-4o, and Qwen-2.5-VL-7B in MathVerse.

## Chain-of-Thought (CoT) Errors

**Question:** Can you explain how line GI and line OS in the image are related to each other?

*A.* parallel
*B.* perpendicular
*C.* on
*D.* incircle

*Ground Truth: A*

**GPT-4o Answer**

Step1: The lines GI and OS are both chords of the circle.
Step2: They are not parallel or perpendicular to each other, nor are they tangent or secant lines.
Step3: Therefore, the correct choice is C: on

## Chain-of-Thought (CoT) Errors

**Question:** Please ignore any shapes formed by red auxiliary lines. Can you identify the type of object CJLNR in the picture?

*A.* isosceles trapezoid
*B.* equilateral quadrilateral
*C.* pentagon
*D.* quadrangle

*Ground Truth: C*

**GPT-4o Answer**

Step1: The object labeled CJLNR in the picture is a quadrangle, which is a four-sided polygon.
Step2: The red auxiliary lines do not form any additional sides or angles that would change the classification of the shape.
Step3: Therefore, the correct answer to the question is D: quadrangle.

Figure 19: Examples of Chain-of-Thought (CoT) Errors in GPT-4o.

## Chain-of-Thought (CoT) Errors

**Question:** What is the shape of object IPYR in the image?

*A.* right trapezoid
*B.* isosceles trapezoid
*C.* equilateral quadrilateral
*D.* equilateral triangle

*Ground Truth: B*

**GPT-4o Answer**

Step1: To determine the shape of object IPYR, we need to analyze the angles and sides.
Step2: The angles at I and P appear to be right angles, and the sides IP and PR are parallel.
Step3: This suggests that IPYR is a right trapezoid. So, the answer is A: right trapezoid.

Figure 20: An example of Chain-of-Thought (CoT) Errors in GPT-4o.

## Chain-of-Thought (CoT) Errors

**Question:** Can you identify the type of object in the picture?

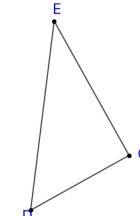

*A.* isosceles triangle
*B.* circle
*C.* scalene triangle
*D.* right triangle

*Ground Truth: D*

**Qwen2.5-VL-7B Answer**

Step1: The image shows a triangle with three vertices labeled as D, C, and E.
Step2: To determine the type of triangle, we need to look at the lengths of its sides.
Step3: If two sides are equal, it is an isosceles triangle. If all sides are of different lengths, it is a scalene triangle. If one angle is 90 degrees, it is a right triangle.
Step4: From the image, we can see that the lengths of the sides DE, EC, and DC are not equal, so the triangle is not isosceles or right.
Step5: Therefore, the correct answer is C: scalene triangle

Figure 21: An example of Chain-of-Thought (CoT) Errors in Qwen2.5-VL-7B.

## Chain-of-Thought (CoT) Errors

**Question:** Can you count the total number of objects in the image?

*A. 3*
*B. 4*
*C. 2*
*D. 5*

*Ground Truth: A*

**Qwen2.5-VL-7B Answer**

Let's count the objects in the image step by step:
Step1: There are 5 points labeled: A, J, G, L, and S.
Step2: There are 4 lines connecting these points: AJ, AG, JL, and LS.
Step3: So, the total number of objects in the image is 5 points and 4 lines, which makes it 9 objects in total.
Step4: However, the options provided are A:3, B:4, C:2, and D:5. It seems there might be a misunderstanding in the question or the options. Based on the image, the correct answer should be: D:5.

Figure 22: An example of Chain-of-Thought (CoT) Errors in Qwen2.5-VL-7B.

## Chain-of-Thought (CoT) Errors

**Question:** Can you explain how line GI and line OS in the image are related to each other?

*A.* parallel
*B.* perpendicular
*C.* on
*D.* incircle

*Ground Truth: A*

**Qwen2.5-VL-7B Answer**

To determine the relationship between line GI and line OS in the image, let's analyze the given options:
Step1: A: Parallel - Lines that never intersect and are always the same distance apart. B: Perpendicular - Lines that intersect at a 90-degree angle. C: On - One line is on the other line. D: Incircle - A circle that is tangent to all three sides of a triangle.
Step2: In the image, line GI and line OS are not parallel, not perpendicular, and not on each other.
Step3: However, they are both radii of the same circle, which means they are equal in length and emanate from the same center point (O).
Step4: Given the options, the correct answer is: C: On

Figure 23: An example of Chain-of-Thought (CoT) Errors in Qwen2.5-VL-7B.

**Recognition Errors**

**Question:** How many objects are there in the image?

A. 1
B. 5
C. 3
D. 2

*Ground Truth: D*

**GPT-4o Answer**

C: 3

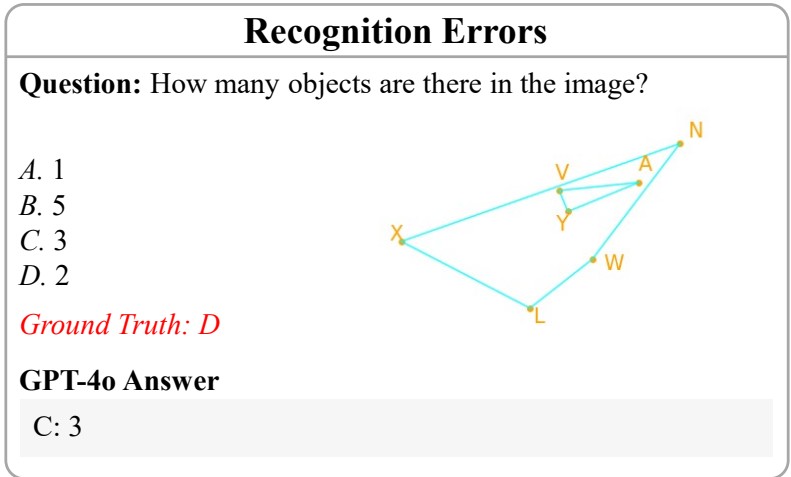

**Recognition Errors**

**Question:** You can only see 2 objects in the picture, can't you?

A. Yes
B. No

*Ground Truth: B*

**GPT-4o Answer**

A: Yes

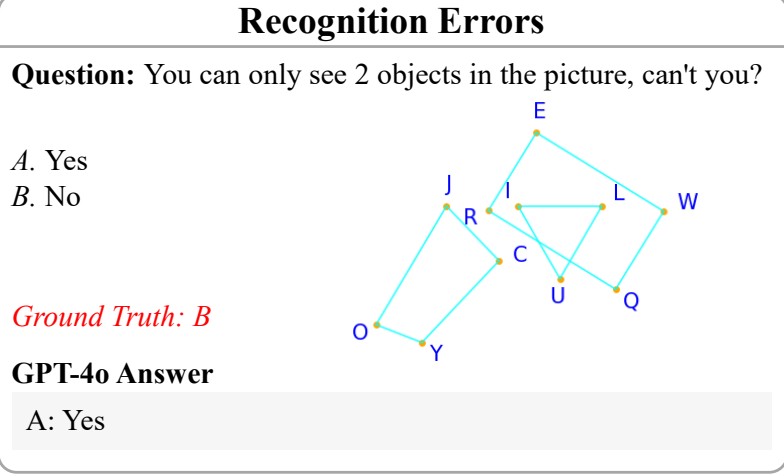

Figure 24: Examples of Recognition Errors in GPT-4o.

**Recognition Errors**

**Question:** Can you identify the type of object QRUW in the picture?

*A.* isosceles trapezoid
*B.* trapezoid
*C.* pentagon
*D.* right trapezoid

*Ground Truth: B*

**GPT-4o Answer**

I can't identify the object QRUW in the picture.

**Recognition Errors**

**Question:** Please provide the bounding box coordinate of the region this sentence describes: right triangle FGK.

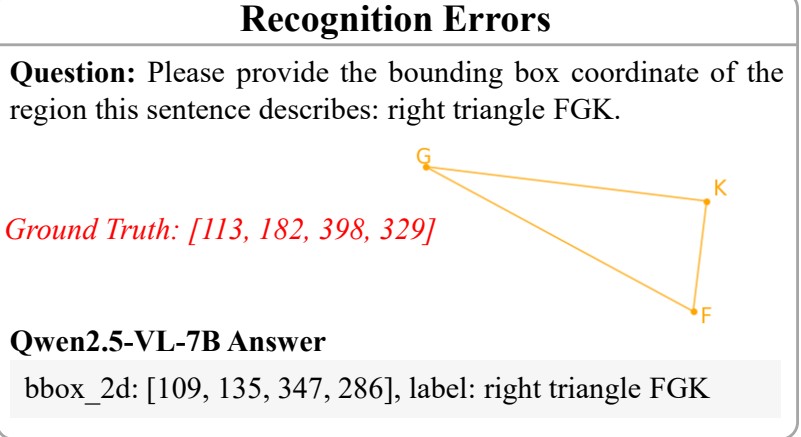

*Ground Truth: [113, 182, 398, 329]*

**Qwen2.5-VL-7B Answer**

bbox_2d: [109, 135, 347, 286], label: right triangle FGK

Figure 25: Examples of Recognition Errors in GPT-4o (top) and Qwen2.5-VL-7B (bottom).

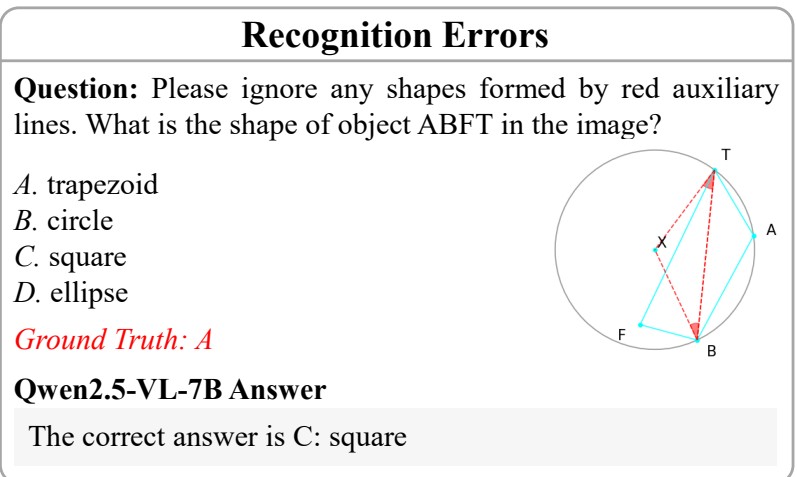

**Recognition Errors**

**Question:** What is the shape of object AFJP in the image?

*A.* isosceles triangle
*B.* equilateral quadrilateral
*C.* rectangle
*D.* right trapezoid

*Ground Truth: C*

**Qwen2.5-VL-7B Answer**

B

**Recognition Errors**

**Question:** Please ignore any shapes formed by red auxiliary lines. What is the shape of object ABFT in the image?

*A.* trapezoid
*B.* circle
*C.* square
*D.* ellipse

*Ground Truth: A*

**Qwen2.5-VL-7B Answer**

The correct answer is C: square

Figure 26: Examples of Recognition Errors in Qwen2.5-VL-7B.

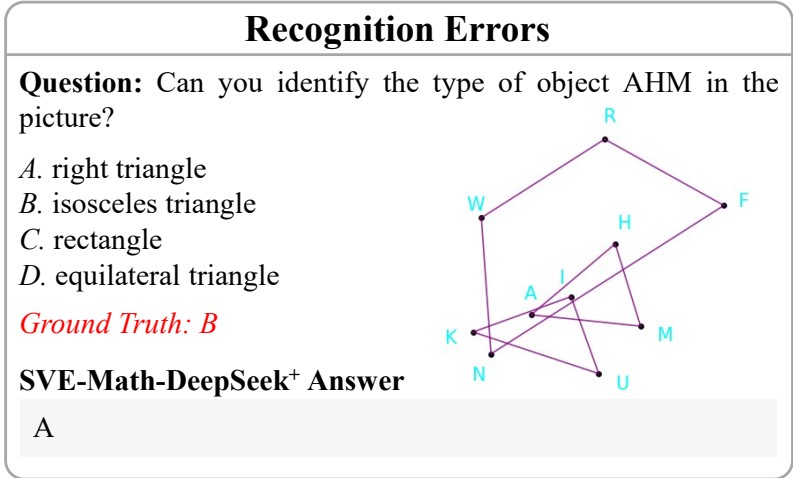

**Recognition Errors**

**Question:** Can you identify the type of object AHM in the picture?

*A.* right triangle
*B.* isosceles triangle
*C.* rectangle
*D.* equilateral triangle

*Ground Truth: B*

**SVE-Math-DeepSeek+ Answer**

A

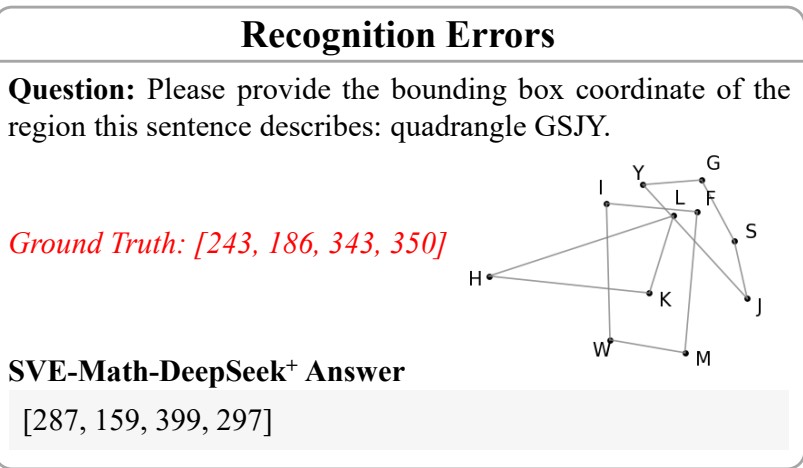

**Recognition Errors**

**Question:** Please provide the bounding box coordinate of the region this sentence describes: quadrangle GSJY.

*Ground Truth: [243, 186, 343, 350]*

**SVE-Math-DeepSeek+ Answer**

[287, 159, 399, 297]

Figure 27: Examples of Recognition Errors in SVE-Math-DeepSeek +-7B.

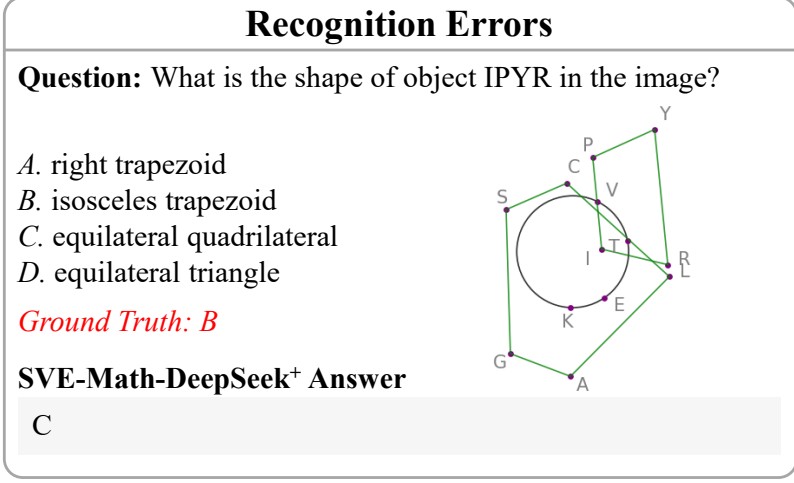

**Recognition Errors**

**Question:** What is the shape of object IPYR in the image?

*A.* right trapezoid
*B.* isosceles trapezoid
*C.* equilateral quadrilateral
*D.* equilateral triangle

*Ground Truth: B*

**SVE-Math-DeepSeek+ Answer**

C

Figure 28: An example of Recognition Errors in SVE-Math-DeepSeek +-7B.

