# OpenReview forum: "Math Blind: Failures in Diagram Understanding Undermine Reasoning in MLLMs"
_ICLR.cc/2026/Conference — ICLR 2026 Poster_

### Official Review · Reviewer_4XJK · 2025-10-26

**Soundness:** 3
**Presentation:** 3
**Contribution:** 3
**Rating:** 6
**Confidence:** 4

**Summary:**

The paper proposes a benchmark called MATHEMETRIC that just tests basic visual understanding, like counting shapes or finding relationships, separate from complex reasoning. They found that even top models fail these simple perception tasks, often just trusting the text blindly. Their fix is a new training dataset, GEOMETRIC, that explicitly teaches models the underlying structure of diagrams. When they trained models on this, perception scores shot up with a 79% gain on grounding. More importantly, this new visual skill transferred, making the models 3-4% better at actual high-level reasoning on other public benchmarks.

**Strengths:**

Primarily, for me, the main strengths of the paper are:

1) Transfer of Learning: The most significant result is the 3-4% improvement on downstream reasoning tasks (Table 2). This demonstrates that fixing the low-level perceptual foundation directly improves high-level reasoning, confirming the paper's central thesis. Achieving this gain without any new reasoning data was a useful finding

2) Isolation of variables: The authors rightly point out that existing benchmarks like MathVista and MathVerse conflate perception and reasoning, making it impossible to diagnose failure. The MATHEMETRIC benchmark is a good diagnostic tool that successfully disentangles these two capabilities.

**Weaknesses:**

A few weaknesses to point out:

1) The paper's solution dataset is synthetic and focuses on plane geometry. While the authors show some cross-domain generalisation to solid geometry and graphs (Table 2), the gains are far smaller. The "graph-based data construction" is not yet extended to these other domains. This makes the solution feel less general than the diagnosis, although the authors mention this in the limitation.

**Questions:**

I have a few required clarifications from the authors:

1) The 3-4% gain on MathVerse -  Could the authors provide more qualitative examples in the appendix showing what kinds of reasoning problems the GEOMETRIC-trained models now solve correctly? Are they all problems that hinge on a single, correctly perceived geometric property, or does the improved perception unlock more complex, multi-step reasoning chains?

2) The text-distractor experiment- While it shows that models over-rely on conflicting text. Did the authors perform the inverse experiment: providing a diagram with clear visual cues (e.g., two parallel lines) but no ambiguous or ambiguous textual information? Do models still fail, or can they reason from the visual-only input? This would further strengthen the "blind faith in text" hypothesis.

---

> ### Author Response · Authors · 2025-11-20
> **Author Response (part1)**
>
> ## We are encouraged by the reviewer’s positive comments and recognition of our main contributions: (1) the MATHEMETRIC benchmark as a diagnostic tool, uncovering the poor diagram perception of current MLLMs, and (2) the potential of perception-to-reasoning transfer capability.
>
> Q1:  The paper's solution dataset is synthetic and focuses on plane geometry. While the authors show some cross-domain generalisation to solid geometry and graphs (Table 2), the gains are far smaller. The "graph-based data construction" is not yet extended to these other domains. This makes the solution feel less general than the diagnosis, although the authors mention this in the limitation.
>
> A1: Thank the reviewer for the thoughtful comment regarding the geometry-specific nature of the current solution dataset. We fully acknowledge this limitation. As noted in Section J of the paper, extending our structured, graph-based data construction beyond plane geometry is an important direction for future work.
>
> While our benchmark provides a general diagnosis across plane geometry, solid geometry, and graphical diagrams (e.g., line/bar/pie charts), our current solution dataset is designed specifically around geometric primitives and spatial relations in planar diagrams. Although we observe improvements in the other domains, they are not yet sufficient. Our planned extensions, outlined in Section J, aim to address this gap.
>
> Developing meaningful node–edge representations for non-geometric diagrams is essential for accurate visual grounding and interpretation. For example:
> (1) In line and bar charts, the underlying structure differs fundamentally from geometric diagrams. Nodes may correspond to data points, axis ticks, group markers, bar segments, or labels, while edges may represent trends, grid alignments, category groupings, or relational mappings across axes.
> (2) In pie charts, nodes correspond to sector boundaries and labels, and edges capture angular divisions and hierarchical grouping.
>
> This is a non-trivial challenge and will require domain-specific structural abstractions, rather than a direct extension of the geometric-primitive framework. We appreciate the reviewer highlighting this point, and we will continue working on developing such domain-specific structural representations.
>
> Q2: The 3-4\% gain on MathVerse - Could the authors provide more qualitative examples in the appendix showing what kinds of reasoning problems the GEOMETRIC-trained models now solve correctly? Are they all problems that hinge on a single, correctly perceived geometric property, or does the improved perception unlock more complex, multi-step reasoning chains?
>
> A2: We thank the reviewer for this excellent question. We provided qualitative inference comparisons between the base model and the GEOMETRIC-fine-tuned models on MathVerse in Figures 16–18. Below, we summarize the key observations:
>
> (1) GEOMETRIC establishes a more reliable perceptual foundation for reasoning.
>
> Across many examples, the base model fails due to an incorrect perceptual premise (e.g., misidentifying the key segment or angle), which causes the reasoning chain to collapse. GEOMETRIC corrects initial perceptual errors, enabling the model to begin its reasoning from correct visual foundations.
>
> For instance, in Figure 16(a), the base model fails to recognize the hypotenuse, while the GEOMETRIC model identifies it and naturally applies the Pythagorean theorem to reach the correct answer.
> Similarly, in Figure 16(b), the GEOMETRIC model correctly perceives the diameter, infers the presence of a right triangle, and applies the appropriate theorem.
>
> Moreover, with improved perception, the model often produces more direct reasoning steps instead of overthinking or searching for unnecessarily complex solutions. Enhanced perception removes ambiguity, allowing the model to commit earlier to the correct reasoning strategy.
>
>
> (2) Perception corrections unlock multi-step geometric reasoning.
>
> In more complex examples, GEOMETRIC improves the model’s ability to execute multi-step reasoning reliably, because the intermediate perceptual relations are now accurate.
>
> For example, in Figure 18(a), both the base model and ours know the relevant theorem—the Inscribed Angle Theorem. However, the base model misperceives key visual cues (e.g., tangent lines), leading to hallucinated angle relations and an incorrect final answer. The GEOMETRIC model correctly perceives the tangency and the right-angle structure, enabling it to execute the theorem correctly and arrive at the correct answer.
>
> **Summary:** Based on our analysis of model responses, although GEOMETRIC may not have the ability to create new logical skills—since it provides no reasoning-style supervision (no CoTs)—it corrects many failures by fixing a single crucial perceptual error, and the resulting improved perception stabilizes multi-step reasoning chains.
>
> We have added the detailed explanation in the appendix of the revision (lines 1231-1238).

---

> ### Author Response · Authors · 2025-11-20
> **Author Response (part2)**
>
> Q3: The text-distractor experiment- While it shows that models over-rely on conflicting text. Did the authors perform the inverse experiment: providing a diagram with clear visual cues (e.g., two parallel lines) but no ambiguous or ambiguous textual information? Do models still fail, or can they reason from the visual-only input? This would further strengthen the "blind faith in text" hypothesis.
>
> A3:  We apologize for any confusion about the setup of text-distractor experiments. To justify the text-biased nature of MLLMs in diagram perception, we conducted two controlled experiments: (1) diagram with clear visual cues and without ambiguous textual information (results shown in Table 4, last panel, Conflicts-w/o), and (2) diagram with the same clear visual cues but with added textual distractors during inference (results shown in Table 4, last panel, Conflicts-w). For accuracy, we use the same measurement (top-1 accuracy): if the model’s answer matches the ground truth, we count it as correct.
>
> Thus, setting (1) corresponds to what the reviewer mentioned—providing a diagram with clear visual cues (e.g., two parallel lines) but no ambiguous textual information. Setting (2) corresponds to providing the same diagram (e.g., two parallel lines) but adding interfering textual information (e.g., “the two lines are perpendicular”). If the model still answers “parallel,” this indicates no text-bias issue. However, as shown by the comparison between the w/o and w results, there is a large gap between the two settings. After prompting with conflicting textual descriptions, current MLLMs show a 22–42\% accuracy drop, even though the diagram remains unchanged. This demonstrates that the models suffer significantly from blind faith in text, despite having clear visual evidence.
>
> We conducted  additional ablations  by explicitly prompting the model with modality-priority instructions. Please refer to our response to Q6 for Reviewer DjMf for the detailed results and discussion.

---

> > ### Comment · Reviewer_4XJK · 2025-11-28
> >
> > I thank the reviewers from addressing my concerns. I apologize from having missed the visual cues setting in the text distractor experiments, and having gone through the other rebuttals by the authors, I find the experiment outcome justified. Moreover the qualitative examples enhance clarity in the understanding of 3-4% gain in MathVerse. I am changing my scores for soundness to 4 and my overall score to 8. Also, I would suggest to extend the work to real world scenarios in the future.

---

> > > ### Author Response · Authors · 2025-11-28
> > >
> > > Thank reviewer for your follow-up feedback and raising the score. We are delighted that we were able to address your concerns and clarify the reasoning improvements. We have added the details of the text-distractor experiment to the revised version. We appreciate your suggestions and will explore real-world extensions in future work.
> > >
> > > Thank you again for your time.

---

### Official Review · Reviewer_5MW7 · 2025-10-28

**Soundness:** 2
**Presentation:** 2
**Contribution:** 2
**Rating:** 4
**Confidence:** 4

**Summary:**

This paper investigates whether the failures of current Multimodal Large Language Models (MLLMs) in mathematical reasoning stem from limited diagram perception. The authors introduce MATHEMETRIC, a benchmark designed to isolate perceptual abilities (shape classification, object counting, relationship identification, grounding) from reasoning, and a synthetic GEOMETRIC dataset that encodes diagram structure as graphs of shapes and relations. Experiments on 18 MLLMs demonstrate that most models perform poorly on basic perceptual tasks, and that improved perceptual training can yield modest reasoning gains across MathVerse, MathVista, and GeoQA.

**Strengths:**

1. **Timely and relevant topic**: The paper addresses an important and underexplored aspect of multimodal reasoning: diagram perception, aligned with ongoing interest in evaluating the visual grounding of MLLMs.

2. **Benchmark contribution**: The proposed MATHEMETRIC benchmark provides a systematic framework to separate low-level perception from reasoning, which is valuable for diagnosing model bottlenecks.

3. **Synthetic dataset design**: The GEOMETRIC dataset introduces structured, graph-like representations of diagrams that could facilitate learning of spatial and relational structures.

4. **Comprehensive evaluation**: The study covers a broad set of models and tasks, including both open- and closed-source MLLMs, and provides ablation studies on factors such as distractors, noise, and text–image conflicts.

**Weaknesses:**

1. **Lack of validation for benchmark annotation quality.**
Since MATHEMETRIC and GEOMETRIC are key contributions, the paper should provide validation or human verification to ensure the accuracy and reliability of the annotated data. Without such validation, it is difficult to assess whether the benchmark truly measures perceptual ability rather than artifacts of synthetic generation.

2. **Unclear causal interpretation of findings.**
The central question "Do current MLLMs genuinely perceive mathematical diagrams?" is not convincingly answered. The reported evidence (e.g., reliance on text, pattern memorization, robustness to distractors) does not establish causal links between perception and reasoning. The argument that stronger perception leads to better reasoning remains correlational, lacking controlled experimental design or counterfactual analysis.

3. **Ambiguity in defining "genuine perception".**
Claims about "blind faith in text" or "Math Blindness" are conceptually interesting but not rigorously operationalized. The study could benefit from clearer definitions or measurable criteria for "genuine diagram perception".

4. **Potential shortcut effects.**
If improved perception data lead to higher reasoning accuracy, it remains unclear whether this reflects better perceptual grounding or merely learning shortcuts in synthetic data. This ambiguity weakens the interpretation of performance improvements as evidence of genuine understanding.

5. **Writing and organization issues.**
The writing could be more concise and polished. Certain sections (e.g., section 3.3) contain excessive detail that obscures the main findings, while others (e.g., motivation and discussion) could better connect empirical results to claims.

**Questions:**

See Weakness

---

> ### Author Response · Authors · 2025-11-20
> **Author Response (part1)**
>
> ## We thank the reviewer for recognizing that diagram perception is an underexplored aspect of MLLMs, and our work contributes a systematic evaluation benchmark as well as synthetic dataset engine to advance research in this area.
>
> Q1: Lack of validation for benchmark annotation quality. Since MATHEMETRIC and GEOMETRIC are key contributions, the paper should provide validation or human verification to ensure the accuracy and reliability of the annotated data. Without such validation, it is difficult to assess whether the benchmark truly measures perceptual ability rather than artifacts of synthetic generation.
>
> A1:  We appreciate the reviewer’s concern regarding annotation quality. Our synthetic data engine guarantees mathematical correctness through formal logic verification rather than random generation, ensuring that both the benchmark and training samples measure true perceptual ability, not artifacts of synthetic construction.
>
> Unlike typical synthetic pipelines that randomly place geometric primitives on a canvas, our approach uses geometric clauses [Trinh et al., AlphaGeometry] as the fundamental units for generating mathematically valid planar geometric figures (see Section F.0.1). Each sampled primitive or set of primitives is passed through a verifier module that checks whether the resulting configuration satisfies fundamental geometric axioms and prerequisite conditions. Only primitives that satisfy these constraints are accepted; otherwise, they are filtered out. For example, parallel lines must not intersect; angle trisection requires sufficient prerequisite points to define the base angle; circle tangency must satisfy the rule that the tangent line is perpendicular to the radius at the point of contact; collinearity requires that all points share a consistent slope and lie on the same line.
>
> This pipeline ensures that every annotated relationship and primitive is mathematically sound, effectively avoiding annotation noise or artifacts. This rigorous construction process is also recognized by *Reviewer Y1aj*, who noted that 'the benchmark is curated with high quality, and the labels are generated in an accurate way (not like some work that use LLMs for sample generation/annotations)'.
>
> Moreover, as acknowledged in lines 209–211 of the main paper, we conduct a comprehensive review to verify answer accuracy, ensure consistency between questions and diagrams, and confirm relevance to the four perception tasks, ensuring high-quality and precise dataset annotations.
>
> Q2: Unclear causal interpretation of findings. The central question "Do current MLLMs genuinely perceive mathematical diagrams?" is not convincingly answered. The reported evidence (e.g., reliance on text, pattern memorization, robustness to distractors) does not establish causal links between perception and reasoning. The argument that stronger perception leads to better reasoning remains correlational, lacking controlled experimental design or counterfactual analysis.
>
> A2: We thank the reviewer for raising this point, though we respectfully disagree that our central question is not convincingly addressed.
>
> (1) On the question: “Do current MLLMs genuinely perceive mathematical diagrams?”
>
> We believe our results provide direct and systematic evidence that current MLLMs do not genuinely perceive symbolic diagrams. We evaluated 20 state-of-the-art MLLMs on MATHEMETRIC, and even the strongest models (GPT-4o, o4-mini-high) achieve below ~50\% average accuracy; for fine-grained grounding—which requires actual spatial localization—most models perform near zero;
> perception ability shows no positive scaling trend: increasing Qwen2-VL from 7B → 72B only improves by 8.3\%.
>
> (2) Clarification about the factors analyzed (text-bias, distractors, robustness tests):
>
> We apologize if the presentation caused confusion.  However, as clearly stated in Section 3.3.1, these ablations were not intended to establish links with reasoning.  Their purpose is to: provide deep diagnostic evidence that MLLMs fail to perceive symbolic diagrams; show that models are vulnerable to small perturbations; highlight that diagram perception is an overlooked challenge (as noted by *Reviewer DjMf’s comment*).
>
> (3) Lacking controlled experimental design for “causal links” between perception and reasoning:
>
> We have designed controlled experiments to demonstrate that enhanced perception directly supports more faithful downstream reasoning.  As shown in Figure 2(b) and Table 7, models trained on GEOMETRIC (our structured perception dataset) achieve substantial perception gains, and consistent +3–4\% improvements on MathVerse without adding any reasoning data. In contrast, models trained on AutoGeo or MAVIS—which also provide diagram-text pairs—obtain similar reasoning performance to the base model, showing no perceptual transfer.  As highlighted  by *three other reviewers*, this finding/transfer is useful and nontrivial.

---

> ### Author Response · Authors · 2025-11-20
> **Author Response (part2)**
>
> Q3: Ambiguity in defining "genuine perception". Claims about "blind faith in text" or "Math Blindness" are conceptually interesting but not rigorously operationalized. The study could benefit from clearer definitions or measurable criteria for "genuine diagram perception".
>
> A3: We appreciate the reviewer’s comment. For how we operationalize “blind faith in text,” we have provided ablation experiments in Table 4 and analysis in lines 431–464. Please also see *our response to Q3 for Reviewer 4XJK* for a more detailed explanation.
>
> Definition of “genuine diagram perception": Genuine perception refers to correctly interpreting the visual structure of a diagram itself, independent of textual hints or shortcuts. To further address the reviewer’s concern, we clarify why diagram perception poses unique challenges for MLLMs. Unlike natural images (bitmaps), symbolic and structured diagrams lack semantic textures or contextual biases (see lines 44–50). Natural-image benchmarks often allow texture-based shortcuts (e.g., cow ↔ grass; Geirhos et al., ICLR 2019). In contrast, diagrams composed of abstract geometric primitives (vectors) require precise interpretation of symbolic meaning and spatial relationships. Distinguishing visually similar shapes (e.g., rectangles vs. trapezoids) therefore requires precise diagram understanding and localization ability, rather than reliance on semantic shortcuts.
>
> Q4. Potential shortcut effects. If improved perception data lead to higher reasoning accuracy, it remains unclear whether this reflects better perceptual grounding or merely learning shortcuts in synthetic data. This ambiguity weakens the interpretation of performance improvements as evidence of genuine understanding.
>
> A4: We thank the reviewer for raising this concern. As noted, our model is trained on synthetic data; however, the performance gains on real-world samples serve as evidence that the model is not simply learning shortcuts. As reported in Table 2 and Table 7/8, models trained with GEOMETRIC improve not only on MATHEMETRIC but also on three external reasoning suites—MathVista, MathVerse, and GeoQA—which contain diverse real diagrams and do not share the rendering patterns of GEOMETRIC.
>
> For the transferable phenomenon, please refer to the qualitative inference comparisons between the base model and the GEOMETRIC-fine-tuned model on MathVerse in Figures 16–18. GEOMETRIC corrects many failures by fixing a single crucial perceptual error, and the resulting improved perception stabilizes the multi-step reasoning chains.
>
> Q5:  Writing and organization issues. The writing could be more concise and polished. Certain sections (e.g., section 3.3) contain excessive detail that obscures the main findings, while others (e.g., motivation and discussion) could better connect empirical results to claims.
>
> A5: We thank the reviewer for this comment. Regarding Section 3.3, this part contains the ablation studies and analyses that directly support our main findings, which is why it includes more detailed investigation. For the connection between empirical results and claims, we have already added cross-references throughout the main paper. For example, in line 86, we cite Figure 6 to support the statement that unnecessarily long chains-of-thought and irrelevant visual content can harm perception accuracy. In line 92, we reference the experimental results in Section 3.3 to support our key observations. For the analysis of “Does stronger perceptual ability lead to better reasoning performance?”, we consistently anchor the discussion with experimental evidence such as Figure 1, Figure 2(a), and Figure 2(b).

---

> > ### Comment · Reviewer_5MW7 · 2025-11-27
> > **Thanks for the explanation**
> >
> > I appreciate the response from the authors. My main concerns are addressed with the additional explanation. The details are a bit delicate so I would suggest that the authors could put them in the paper and improve the presentation in the next version.
> >
> > With such clear statements in the paper, I think the work is ready for the publication. Therefore, I have updated my score accordingly: soundness 2->3, contribution 2->3, and overall score 4->6.

---

> > > ### Author Response · Authors · 2025-11-28
> > >
> > > Thank reviewer for the follow-up and for updating your scores. We appreciate the reviewer’s careful reading and guidance throughout the process. These additions will be added to the final version.
> > >
> > > Thank you again for your time.

---

### Official Review · Reviewer_DjMf · 2025-10-30

**Soundness:** 3
**Presentation:** 3
**Contribution:** 3
**Rating:** 6
**Confidence:** 4

**Summary:**

This paper investigates a critical but underexplored limitation of Multimodal Large Language Models (MLLMs): their inability to perceive mathematical diagrams correctly, leading to reasoning failures. The authors introduce two main contributions: 1. MATHEMETRIC, a diagnostic benchmark that isolates low-level perception (shape classification, counting, relationship identification, grounding) from high-level reasoning across three domains—plane geometry, solid geometry, and graphs. 2. GEOMETRIC, a structured training dataset encoding diagrams as graphs of geometric primitives with attributes and relationships.

Through evaluations of 18 MLLMs (generic and math-specific), the paper demonstrates that current models fail severely in fine-grained perception (e.g., <20% in grounding) but that training on GEOMETRIC yields substantial perception gains (+79% grounding accuracy) and modest reasoning improvements (+3–4% on MathVerse/MathVista). Finally, the authors argue that perception is a fundamental bottleneck for reasoning and that structure-aware visual pretraining can partially bridge this gap.

**Strengths:**

The work introduces a novel and valuable perspective on multimodal reasoning by isolating diagram perception from mathematical reasoning, a distinction often blurred in existing benchmarks like MathVista and MathVerse. The proposed “Math Blind” hypothesis is conceptually fresh and empirically grounded, offering a clear diagnostic lens for assessing visual understanding in symbolic domains. The paper is clearly written, visually well-presented, and effectively highlights a long-overlooked challenge in MLLMs—the perception of structured, symbolic diagrams.

**Weaknesses:**

- The paper operationalizes perception through four basic CV-style tasks (classification, counting, grounding, relationship). This oversimplifies the rich perceptual hierarchy involved in diagram understanding, which may include topological reasoning, implicit constraints (e.g., angle or symmetry inference), and multi-object composition.
- The claimed perception-to-reasoning transfer (+3–4%) is small compared to the massive perception improvement (+79%), weakening the causal claim that “low-level perception is the main bottleneck of reasoning.”
- Recent high-performing multimodal reasoning models such as Vision-R1, MINT-CoT [Chen et al., 2025], and the MathVision benchmark [Wang et al., 2024] are not included.
- The “blind faith in text” phenomenon is intriguing, yet the paper lacks controlled prompts that explicitly modulate text–image priority (e.g., “prefer textual information” vs. “prefer visual cues”). Without this, the analysis remains observational, not causal.

**Questions:**

1. How would your findings change under interleaved visual token reasoning paradigms (e.g., MINT-CoT [1]) where visual tokens are mixed directly with text during CoT generation? Could these newer paradigms render the perception–reasoning separation less meaningful?

2. When textual and visual information conflict, how do you determine which should be “correct”? Have you tried prompting models with explicit modality preference instructions to measure alignment sensitivity in the relative experiments (“blind faith in text”)?

3. The inclusion of recently released reasoning-enhanced models, such as Vision-R1 (Huang et al., 2025), needs to be added to strengthen the study's central claim regarding perception as the primary bottleneck.

4. To enhance the evaluation, it is recommended to incorporate advanced multimodal math benchmarks like MathVision [3].


References:
 [1] Chen et al., MINT-CoT: Enabling Interleaved Visual Tokens in Mathematical Chain-of-Thought Reasoning, arXiv:2506.05331, 2025.
 [2] Huang et al., Vision-R1: Incentivizing Reasoning Capability in Multimodal Large Language Models, arXiv:2503.06749, 2025.
 [3] Wang et al., Measuring Multimodal Mathematical Reasoning with MathVision Dataset, NeurIPS 2024.

---

> ### Author Response · Authors · 2025-11-20
> **Author Response (part1)**
>
> ## Thank you for the reviewer’s insightful comment. We are excited that the reviewer agrees with our key insight that the perception of structured, symbolic diagrams is a long-overlooked challenge for MLLMs. Our proposed “Math Blind” hypothesis is conceptually fresh \& empirically grounded, and MATHEMETRIC provides a clear diagnostic lens for assessing visual understanding in symbolic domains.
>
> Q1:  The paper operationalizes perception through four basic CV-style tasks (classification, counting, grounding, relationship). This oversimplifies the rich perceptual hierarchy involved in diagram understanding, which may include topological reasoning, implicit constraints (e.g., angle or symmetry inference), and multi-object composition.
>
> A1: We appreciate the reviewer’s thoughtful observation about the richness and structural complexity of diagram understanding. We agree that diagram interpretation involves higher-level topological reasoning and implicit geometric constraints. Our basic perception tasks capture certain elementary aspects of structural and topological reasoning. For example, shape classification requires determining which points form which line segments and which segments compose a closed shape, while the relationship task (parallelism, perpendicularity, collinearity) evaluates geometric constraints. We acknowledge that this level of topological evaluation is not sufficient to cover the full complexity of diagram understanding.
>
> That said, these basic perception tasks constitute foundational abilities upon which diagram interpretation and logical reasoning necessarily depend. Many richer perceptual phenomena can be decomposed into compositions of the primitive skills we measure. For example, to infer that two sides are symmetric requires: (1) classifying them as line segments, (2) grounding their spatial positions, and (3) extracting their relationship (parallel, equal length). Likewise, multi-object composition depends on correctly identifying constituent shapes, which is reflected in the classification and counting tasks.
>
> We thank the reviewer for this valuable suggestion. Fully evaluating the rich perceptual hierarchy of diagrams requires more sophisticated task designs and metrics, and we consider this an important direction for future work.
>
> Q2: The claimed perception-to-reasoning transfer (+3–4\%) is small compared to the massive perception improvement (+79\%), weakening the causal claim that “low-level perception is the main bottleneck of reasoning.”
>
> A2: We thank the reviewer for raising this important point. We would like to clarify that the relative gains on perceptual MATHEMETRIC and reasoning benchmarks would not be interpreted as a weak relationship between the two. However, if the reviewer feels that the “main bottleneck” claim is too broad in the context of our paper, we are happy to revise the phrasing, as follows:
>
> - Abstract, lines 30-31: Our findings demonstrate that low-level perception is fundamental to high-level reasoning in mathematical MLLMs-> Our findings demonstrate that low-level perception supports faithful high-level reasoning in mathematical MLLMs.
>
> - Introduction, line 138: underscore perception as the key bottleneck -> showing that structure-aware geometric samples provide essential visual understanding for accurate reasoning
>
> Regarding the difference between the +79\% perception improvement and the +3–4\% reasoning improvement, this arises from two factors:  1) GEOMETRIC is designed specifically to improve diagram perception, and MATHEMETRIC directly evaluates this ability; 2) general reasoning benchmarks such as MathVerse and MathVista require substantial high-level textual CoT reasoning, which GEOMETRIC does not target, as it contains no reasoning-style supervision. After evaluating current MLLMs on our diagnostic test suite and conducting comprehensive ablation studies, the main claims of our work are:  (1) current MLLMs lack the ability to reliably and faithfully perceive diagrams, and (2) enhancing perception ability can support faithful downstream reasoning.
>
>
> In terms of perception-to-reasoning transfer, although we do not introduce any specialized mechanism to couple perception with CoT reasoning, we still observe transfer phenomena: models begin to correctly identify geometric relations, angle relations, and relational cues that support reasoning (Figures 16–18). For instance, in Figure 16 (a), the model aligns the visual cues (e.g., the hypotenuse and perpendicular) with the Pythagorean theorem. Overall, our paper primarily provides a solution for enhancing diagram perception in MLLMs, and even without any explicit transfer mechanisms, the reasoning performance still improves by +3–4\%. How to explicitly integrate enhanced perception into the reasoning pipeline is an important and promising direction for future work (as noted in lines 1272–1275, Appendix). As the reviewer noted, MINT-CoT is closely related.  We further discuss in the next response.

---

> ### Author Response · Authors · 2025-11-20
> **Author Response (part2)**
>
> Q3: Relevance of MINT-CoT and Vision-R1 to our investigations:
>
> A3-1: We appreciate the reviewer’s suggestion to contextualize our work with recent advances in perception–reasoning integration such as MINT-CoT and Vision-R1. These approaches align with our findings that perception is critical for reasoning, yet operate at different scopes.
>
>
> *Relation to MINT-CoT.* MINT-CoT explicitly injects interleaved visual tokens into the chain-of-thought, enabling the model to reference visual evidence during reasoning. This represents an important direction for coupling perception and reasoning step-wisely. In contrast, our work focuses on improving diagram perception itself, without introducing any specialized mechanism for mixing visual and textual tokens in the reasoning process. As a result, the two lines of work are compatible: MINT-CoT studies how visual information is integrated into CoT reasoning; our work studies whether the model can accurately perceive symbolic diagrams in the first place, and how enhanced perception naturally transfers to reasoning.
>
>
> *Relation to Vision-R1.* Vision-R1 aims to incentivize reasoning ability on top of a strong base model using reinforcement learning. A key factor in Vision-R1’s success is the quality and diversity of its cold-start data: over 43 curated datasets spanning mathematical diagrams, science and medical figures, general QA images, and figure-understanding datasets. Importantly, during data construction, Vision-R1 generates pseudo-CoT using image captions, giving the model holistic visual context that supports the development of high-quality reasoning traces. While both works support the idea that perception is a prerequisite for effective multimodal reasoning, Vision-R1 focuses primarily on improving reasoning ability, whereas our work focuses on explicitly enhancing symbolic diagram perception. **This distinction is crucial: reasoning improvements in Vision-R1 rely heavily on RL strategies and large-scale, high-quality CoT training samples, not on explicit perceptual enhancement.**
>
> | Model              | MATHGLANCE_plane | MATHGLANCE_solid | MATHGLANCE_graphs | MathVista | MathVerse | GeoQA |
> |--------------------|------------------|------------------|-------------------|-----------|-----------|-------|
> | Qwen2.5-VL-7B      | 44.0             | 69.0             | 65.7              | 68.2      | 49.2      | 76.4  |
> | Vision-R1-7B       | 39.6             | 66.6             | 68.4              | 73.5      | 52.4      | 78.9  |
> | MINT-CoT-SFT       | 39.1             | 61.7             | 63.3              | 67.8      | —         | 62.1  |
> | Qwen2.5-VL-7B$^{+}$ (ours) | 78.5             | 71.9             | 68.2              | 70.3      | 52.8      | 79.6  |
>
> **Summary:**
>  - Our work: focuses on improving diagram perception and demonstrates measurable transfer from enhanced perception to downstream reasoning.
> - MINT-CoT: interleaves visual tokens within CoT to produce more visually aligned reasoning.
> - Vision-R1: incentivizes reasoning ability through reinforcement learning, built on high-quality, large-scale perception–reasoning training data.
>
> **We added this contextualization in the revised version (Section C) to clarify how our investigation fits within the broader landscape of perception–reasoning research.**

---

> ### Author Response · Authors · 2025-11-20
> **Author Response (part3)**
>
> A 3-2:  After evaluating Vision-R1 on MATHEMETRIC (above table), we observed pronounced overthinking issues, and we therefore analyze them in depth here.
>
>
> (1) Vision-R1 addresses overthinking in reasoning tasks using Progressive Thinking Suppression Training to control thought length. However, despite these improvements, Vision-R1 still suffers from substantial overthinking on our perception benchmark: under the same settings, inference on MATHEMETRIC takes 32.9 minutes for our model but around 68 hours for Vision-R1. In particular, Vision-R1 tends to produce long, full-diagram descriptions instead of completing the required perception task, and frequently triggers repetitive ''Hmm, maybe…'' or ''Wait, no...'' throughout its response. This further illustrates that perception-oriented tasks require efficient and accurate visual grounding, which Vision-R1 does not explicitly target.
>
> (2) Integrating explicit perceptual ability into RL-based reasoning frameworks is a promising future direction, but doing so would require task-specific reward functions for perception policy learning, as suggested in recent work such as Perception-R1 (En Yu et al.). This cannot be achieved simply through a unified next-token prediction loss in standard supervised fine-tuning (SFT), and fully exploring such RL strategies is outside the scope of the present work. We leave this as an important direction for future research, where our structured perception-training dataset could serve as a valuable foundation for perception policy reinforcement learning on diagrammatic tasks.
>
> (3) Overthinking neither improves simple perception accuracy nor uses computation efficiently. We fine-tune using only 200K samples (100K structured diagram–caption pairs + 100K conversational instruction data), enabling direct-answer perception and step-by-step reasoning for higher-level geometric problems. The synthetic dataset is GPU-free to generate, and  our fine-tuning runs on 8× A100 GPUs (7B) or 16× H100 GPUs (32B) within less 16 hours. In contrast, Vision-R1 requires heavier computation. According to the authors, training data processing requires 128× NVIDIA H800 (80GB) GPUs for ~2 days. Its training further requires 64× H800 GPUs over a large-scale dataset spanning diverse domains.
>
> Q4:  Inclusion of recently released reasoning-enhanced models, such as Vision-R1 and MINT-CoT on MATHEMETRIC, also incorporating the advanced multimodal math benchmark MathVision.
>
> A4: We have added the MATHEMETRIC evaluation results of Vision-R1 and MINT-CoT in Table 1, and included our performance on the MathVision benchmark in Table 2.
>
> For MATHEMETRIC evaluation, please refer to the table in our response to Q3. The table below reports our model’s evaluations on the MathVision benchmark.
>
> |                   | Qwen2.5-VL-7B | Qwen2.5-VL-32B | Math-LLaVA-13B | G-LLaVA-7B | MultiMath-7B | SVE-Math-DeepSeek-7B | SVE-Math-DeepSeek-7B^+ | Qwen2.5-VL-7B^+ | Qwen2.5-VL-32B^+ |
> |-------------------|---------------|----------------|----------------|------------|--------------|------------------------|--------------------------|-------------------|--------------------|
> | Math-Vision       | 25.1          | 31.9           | 15.7           | 12.1       |-       | 14.4                   |16.6                   | 27.3             | 33.3               |
>
> Q5:  How would your findings change under interleaved visual token reasoning paradigms (e.g., MINT-CoT [1]) where visual tokens are mixed directly with text during CoT generation? Could these newer paradigms render the perception–reasoning separation less meaningful?
>
> A5: We provide a detailed discussion in our response to Q3. We agree that interleaved visual–token reasoning is a promising formulation for multimodal reasoning. MINT-CoT address the architectural question of 'when to mix visual/text tokens', while our work addresses the fundamental question of 'what capabilities models possess'. These directions are complementary: even with interleaving mechanisms, accurate perception is still required—otherwise the visual tokens become less meaningful and less interpretable.

---

> ### Author Response · Authors · 2025-11-20
> **Author Response (part4)**
>
> Q6: The “blind faith in text” phenomenon is intriguing, yet the paper lacks controlled prompts that explicitly modulate text–image priority (e.g., “prefer textual information” vs. “prefer visual cues”). Without this, the analysis remains observational, not causal. When textual and visual information conflict, how do you determine which should be “correct”?
>
> A6: We thank the reviewer for this constructive suggestion. In the main paper, we used a neutral prompt that did not favor either modality. For the correctness under conflicting information,  we aligned the ground truth with the visual cues in the diagrams, treating the textual information as the source of conflict. To further address the reviewer’s concern about causal interpretation, we conducted additional controlled experiments using system prompts that explicitly modulate modality priority: (1) visual-priority prompt: "Carefully examine the diagram and prioritize visual information. Only use text labels to confirm what you observe visually"; (2) text-priority prompt:
> "Focus on the textual labels and annotations in the diagram. Use the visual structure to support the textual information."
>
>
>   |Task (rlat./Plane) | w/o Conflicts | w/neutral | w/visual-priority | w/text-priority |
>   |-------------------|---------------|-----------|-------------------|-----------------|
>   |Qwen2VL-7B         | 53.0          | 24.5      |      25.5         |   25.5         |
>   |Qwen2.5VL-7B       |  52.0         | 30.0      |       30.5        |     28.33          |
>   |DeepSeek-VL2-Tiny  |     32.0      | 3.5       |   11.0            |      9.5           |
>   |DeepSeek-VL2-Small |     48.5      | 20.5      |  22.0             |       19.5          |
>   |LLaVA-v1.5-13B     | 42.0          | 16.0      |  15.0             |   14.5          |
>   |InternVL2.5-8B     | 60.0          | 18.0      |  17.0             |   15.5          |
>   |GPT-4o             |  62.5         | 38.0      |  39.0             |   34.0          |
>   |SVE-Math-DeepSeek-7B| 51.0         | 15.0      |  15.0             |   12.5          |
>   |SVE-Math-DeepSeek^+-7B|  96.5      | 82.5      |  87.5             |   84.0 |
>
>
> The results show that even when explicitly instructed to prioritize visual information, the model still exhibits blind faith in text, although its performance is marginally better than under the explicit text-priority prompt. **We have included these results (Table 5) and add corresponding discussion in the revised version to strengthen the causal analysis of the “blind faith in text” phenomenon.**

---

### Official Review · Reviewer_Y1aj · 2025-11-04

**Soundness:** 3
**Presentation:** 3
**Contribution:** 4
**Rating:** 6
**Confidence:** 4

**Summary:**

This paper address an important aspect of evaluating MLLMs on visual math problems: Isolating the perception of simple visual concepts from complex mathematical reasoning and problem solving. To this end, the authors present a benchmark that tests basic perception skills like counting, shape recognition, and locating objects, and a training dataset of geometric shapes and relationships to improve the models' perfromance. Their results show while models peform poorly on the benchmark, using the training set their performance improves, and even results in improvements on the other reasoning benchmarks.

**Strengths:**

- This paper addresses an important challenge in evaluating the performance of the MLLMs on visual understanding and conceptualization
- The benchmark is curated with a high quality, the labels are generated in an accurate way (not like some work that use LLMs for sample generation/annotations)
- The paper also provides a training set for training the models
- The results of training using this set are strong, with high improvements on their proposed bencmark, but more interestingly, improvement on other reasoning benchmarks (MathVerse and MathVista)

**Weaknesses:**

- The object grounding task (with. example in Figure 3, rightmost) is not. It is of no surprise that the models are performing so poorly on this task, given the difficult nature of finding the exact pixel coordinates from an image, which is at the same time unnecessary for visual perception and reasoning. A better alternative to this task would have been to ask the models to give the object vertex letters (for instance in the same task: Q: Please provide the List the vertex labels of the object: scalene triangle. A: HVI)
- The shape classification task can have a shortcut for models: If they see a vertex label of three letters, it is a triangle, so even without the image they can response correctly if only one answer in the answer choices is a triangle, or with a higher probability than random (0.25) if not all answer choices are triangles.
- Some of the graphs and figures could improve for clarity. For instance, Figure 1 is hard to compare different models. maybe a radar plot could have been more illustrative. Also the colors of Figure 5 could be chosen better. It's quite hard to follow the light brown and yellow colors.
- While the paper states that the tasks are easy for humans multiple times, there is no human performance baseline provided. Going back to the first point, the grounding task would be impossible for humans only with looking at the images.

I am willing to increase my score if these concerns are addressed.

**Questions:**

- Could you please give some details on the size and samples of the training set (GeoMetric)?
- How do you make sure that the weakness 2 is not happening?
- Your work resembles a recent work on conceptualisation (Visual Graph Arena: Evaluating Visual Conceptualization of Vision and Multimodal Large Language Models, ICML2025). In that works the authors observe two anomalies in perception, the Easier-Worse and  Middle-Score. Did you observe any similar anomalies in your experiments?
- Do you have an idea to evaluate not only the final answer, but also the models' CoT accuracy on the benchmark? (sometimes the models give the correct answers by chance, with the wrong CoT)

---

> ### Author Response · Authors · 2025-11-20
> **Author Response (part1)**
>
> # Thank you for your thoughtful and constructive feedback that help refine our work further. We are encouraged by the recognition that (1) our study highlights the importance of isolating basic perceptual skills, and (2) our proposed structured primitive training data can strengthen both perception ability and high-level reasoning.
>
> Q1: The object grounding task (with. example in Figure 3, rightmost) is not. It is of no surprise that the models are performing so poorly on this task, given the difficult nature of finding the exact pixel coordinates from an image, which is at the same time unnecessary for visual perception and reasoning. A better alternative to this task would have been to ask the models to give the object vertex letters (for instance in the same task: Q: Please provide the List the vertex labels of the object: scalene triangle. A: HVI)
>
> A1: Thank you for the helpful suggestion. We conducted the vertex-list grounding experiment and compared it with numeric-coordinate grounding; the results are shown in the table below. We appreciate this perspective that vertex lists offer an alternative abstraction for planar geometric objects grounding. Furthermore, vertex-list evaluation was incorporated into our counting task, where the model is required to identify the vertex lists corresponding to the queried object (e.g., right triangle-> CDX, ORT; see Figure 9(a), fourth row, in the appendix). We have added this new experimental result in Table 11 of the revised version.
>
> *Why we designed the grounding task with numerical coordinates:*
> Our grounding task follows the standard referring expression grounding paradigm widely used in recent MLLM literature on **natural images**, including LISA (Lai et al., 2023), Ferret (You et al., 2023), and Shikra (Chen et al., 2023). The common task format—“Given an image and a textual reference, return the bounding box location of the mentioned region”—requires more text-guided visual localization than matching pixel coordinates. This grounding ability has been recognized as a prerequisite for higher-level visual reasoning (e.g., TPAMI 2025 survey “Towards Visual Grounding”). Our goal was to adopt the same widely accepted evaluation metrics in diagram-based grounding.
>
> *The poor performance of current MLLMs unsurprising:* We appreciate the reviewer’s observation that precise coordinate prediction is difficult. Indeed, this difficulty is exactly what reveals the unique challenges of diagram perception.
> On natural-image grounding tasks, such as RefCOCO/RefCOCO+/RefCOCOg, models perform reasonably well (e.g., Qwen2.5-VL-3B achieves 88.6 / 81.9 / 85.1). On diagram grounding, however, even larger models perform dramatically worse (e.g., Qwen2.5-VL-7B achieves 18.3 / 0.0 / 3.2 on Plane/Solid/Graph diagrams).
> This reflects a large domain gap between natural images and symbolic diagrams (bitmaps vs. vectors): diagrams are semantically sparse but structurally rich, requiring models to interpret symbolic meaning and spatial relationships precisely. This helps explain why grounding performance on diagrams is substantially lower than on natural images, and it further underscores the necessity of evaluation for diagram grounding.
>
> | Task (grd./Plane)      | Qwen2-VL-7B | Qwen2.5-VL-7B | LLaVA-v1.5-7B | InternVL2.5-38B | GPT-4o | G-LLaVA-VL-7B | MultiMath-7B | SVE-Math-DeepSeek-7B | SVE-Math-DeepSeek+-7B |
> |-----------|-------------|---------------|----------------|------------------|--------|----------------|---------------|------------------------|-------------------------|
> | vertex    | 45.6 |   49.1   | 1.8  | 42.7 |  44.8  |    3.9  |  33.1  | 41.7 | 89.3 |
> | numerics  | 12.8  | 18.5 | 14.2 |  2.5   | 1.1  |  0.4  | 1.1  |  3.6   |  82.9  |
>
> Q2:  The shape classification task can have a shortcut for models: If they see a vertex label of three letters, it is a triangle, so even without the image they can response correctly if only one answer in the answer choices is a triangle, or with a higher probability than random (0.25) if not all answer choices are triangles.
>
> A2-1: We thank the reviewer for highlighting the potential shortcut in shape classification via vertex labels. We avoid such shortcuts by carefully designing hard distractors in the answer choices: in 75\% of the questions, at least one distractor is selected from the same polygon family, ensuring that the model cannot rely solely on vertex-count cues. Specifically, when the ground-truth object has three vertices (e.g., a scalene triangle), the distractors are selected from other triangle types—such as isosceles, right, or equilateral triangles. To avoid confusion and clearly demonstrate this hard-distractor design, we have updated the first sample in Figure 3 accordingly. The corresponding case for four-vertex shapes is already shown in Figure 9 (a) (first row) of the appendix, where a ground-truth **rectangle** is paired with a **right trapezoid** as the distractor.

---

> ### Author Response · Authors · 2025-11-20
> **Author Response (part2)**
>
> A2-2: To further address the reviewer’s concern, we conducted an additional controlled experiment using 200 planar geometric diagrams in which vertex letters were replaced by purely visual numeric markers (e.g., “1, 2, 3, ...”) that do not encode polygon type. For instance, an isosceles triangle is labeled as “1” rather than by a three-letter vertex label. Example cases are provided in Figure 8, and the corresponding ablation results are included in the appendix (Table 12).
>
> | Task (cls./Plane)      | Qwen2-VL-7B | Qwen2.5-VL-7B | LLaVA-v1.5-7B | InternVL2.5-38B | GPT-4o | G-LLaVA-VL-7B | MultiMath-7B | SVE-Math-DeepSeek-7B | SVE-Math-DeepSeek+-7B |
> |------------------------|-------------|---------------|---------------|-----------------|--------|---------------|--------------|----------------------|-----------------------|
> | vertex                 | 50.7      | 48.6           | 34.6        |  55.3          | 53.1  |  31.6       |  44.1       | 43.9               | 72.6                |
> | markers                | 43.3       | 47.9           | 35.3         |  43.3           | 54.8   |  28.8        |  33.2      | 41.1                | 71.7                 |
>
> Based on the results, we observe variation across models when switching the object indicator from vertex labels to visual markers. This shows that our distractor design is sound and that models cannot rely solely on vertex-count cues. If such shortcuts dominated, all models would exhibit a similar large drop when moving to markers—but they do not. Models like Qwen2.5-VL-7B, LLaVA-v1.5-7B, SVE-Math-DeepSeek-7B, and ours maintain consistent performance. These irregularities reflect differing abilities to interpret visual markers: for example, GPT-4o handles markers well, while InternVL2.5-38B performs better with vertex labels.
>
> Q3. Some of the graphs and figures could improve for clarity. For instance, Figure 1 is hard to compare different models. maybe a radar plot could have been more illustrative. Also the colors of Figure 5 could be chosen better. It's quite hard to follow the light brown and yellow colors.
>
> A3: We appreciate the reviewer’s suggestion regarding figure clarity, and we have made the corresponding improvements. Specifically, in Figure 1, we replaced the original visualization with a radar plot to present the aggregated performance differences more clearly. We also adopted a more distinct color palette, replacing the previous light brown and yellow tones with gray and navy blue, to enhance readability throughout the figures, especially in Figure 5 and Figure 6.
>
> Q4:  While the paper states that the tasks are easy for humans multiple times, there is no human performance baseline provided. Going back to the first point, the grounding task would be impossible for humans only with looking at the images.
>
> A4: We thank the reviewer for raising this important point. We previously asked our authors to complete the perception tasks in the benchmark, but due to the page limitations of the main paper, we did not include these results in Table 1. We apologize for this omission, and we have now added the corresponding results in the revised version.
>
> We agree that humans do not produce explicit pixel-level bounding box coordinates when viewing diagrams. However, humans rely on an implicit coordinate system in the visual cortex that supports structured spatial representations. This form of implicit localization is well-established in vision science, and bounding-box coordinates serve only as a measurement proxy for such internal spatial mapping. We also appreciate the reviewer’s suggestion of an alternative abstraction for planar geometry.  For the grounding task in our human evaluation, our authors used LabelImg, a lightweight tool for drawing bounding boxes with automatic coordinate output.
>
> Q5: Could you please give some details on the size and samples of the training set (GeoMetric)?
>
> A5:
> The training set of GeoMetric contains 200K samples: 100K image–caption pairs for projector pre-alignment, followed by 100K instruction-style conversational samples for second-stage SFT. Figures 13–15 illustrate representative examples from these two data formats in GeoMetric. We provide full training details in Section C of the appendix, where we describe the implementation and training procedure for both 7B and 32B models.

---

> ### Author Response · Authors · 2025-11-20
> **Author Response (part3)**
>
> Q6. Your work resembles a recent work on conceptualisation (Visual Graph Arena: Evaluating Visual Conceptualization of Vision and Multimodal Large Language Models, ICML2025). In that works the authors observe two anomalies in perception, the Easier-Worse and Middle-Score. Did you observe any similar anomalies in your experiments?
>
> A6: Thank you for pointing out this important connection to Visual Graph Arena (VGA). Your question motivated us to analyze our results through the lens of this valuable work.
>
> In our experiments, we did not observe the Easier–Worse or Middle-Score anomalies in a systematic manner. As defined in VGA, the Middle-Score Anomaly arises only in atomic tasks where the correct behavior should be either near 100\% (the concept is fully known) or near chance level (the model is guessing). Any “middle” accuracy is therefore considered anomalous. By contrast, the perception tasks in MATHEMETRIC are compositional, not atomic. Their difficulty varies smoothly with factors such as: the number of objects in the diagram; the independence and interaction of geometric primitives and the inherent visual complexity of the configuration.
>
> For example, even if a model fully understands the definition of a triangle—a polygon with three non-collinear points connected by three line segments—the difficulty of classification or counting naturally increases as the diagram becomes more cluttered or contains similar shapes. As a result, it is expected for models to achieve intermediate accuracies (e.g., 40–70\%) rather than clustering at the extremes, and such behavior is not anomalous in our setting.
>
> Regarding the Easier–Worse Anomaly, we provide relevant analyses in Section 3.3.1. Increasing perceptual difficulty through factors such as object count, diagram fidelity, or object quality consistently reduces accuracy in predictable ways.
>
> Overall, MATHEMETRIC does not exhibit the types of anomalies identified in VGA because our benchmark focuses on graded, structured perception, rather than atomic all-or-nothing conceptual tasks. We have added a brief comparison in the revised version to clarify how our compositional perceptual structure differs from the atomic task structure of VGA with respect to the Easier–Worse and Middle-Score anomalies.
>
> **We have added the corresponding discussion of Visual Graph Arena (VGA) in the Related Work section.**
>
> Q7.  Do you have an idea to evaluate not only the final answer, but also the models' CoT accuracy on the benchmark? (sometimes the models give the correct answers by chance, with the wrong CoT).
>
> A7: We appreciate the reviewer’s insightful question about evaluating not only the final answer but also the correctness of the model’s chain-of-thought (CoT)! This is an important and challenging direction for now. In this pilot version of the benchmark, our goal was to measure perception ability under direct-answer (<no thinking>) settings, without expecting the model to generate long reasoning chains. As shown in Figure 6, models that engage in step-by-step CoT often accumulate larger perceptual errors compared with direct answers. This observation aligns with recent findings in Think or Not Think (Ming Li et al., 2025), which show that deliberate thinking may harm low-level perceptual tasks.
>
> To better understand this phenomenon, we manually inspected the intermediate reasoning traces of MLLMs (Figures 19–23). We found that many failures arise from hallucinated visual cues: the model mis-describes the diagram during CoT, leading to incorrect final answers. The reviewer raises an excellent point that models may sometimes reach the correct answer by chance, despite producing an incorrect CoT. Designing a reliable method to verify perception-level CoT is therefore non-trivial. Current work typically uses LLM-as-a-judge to assess reasoning correctness, but this approach remains imperfect because the verifier itself may hallucinate or misinterpret visual details.
>
> Looking forward, we believe a promising direction is to develop structured and rule-based evaluation metrics for visual CoTs. The key challenge is ensuring that the model’s perceptual reasoning is expressed in a verifiable, non-ambiguous format rather than in free-form natural language. For instance, if visual CoT can be constrained to follow the hierarchical structure of geometric primitives—objects → attributes → relations—then it becomes much easier to extract symbolic elements for rule-based evaluation. This idea resonates with emerging work in perception-policy modeling in reinforcement learning, where verifiable and traceable state representations are essential.
>
> We are excited about this direction and plan to explore verifiable perceptual CoTs in future versions of the benchmark.

---

> > ### Comment · Reviewer_Y1aj · 2025-11-25
> >
> > I thank the authors thorough responses and the time they spent on the rebuttal and revision. I especially appreciate the new results provided for the object grounding and the shape classification tasks.  These results address my main concerns, and hence I will raise my score.
> >
> > I have one further question regarding the human performance added to the Table 1.  How did the humans do the "object grounding" tasks, and how was their performance evaluated? Did you ask human subjects to draw the bounding boxes or also provide pixel coordinates? The latter seems to be impossible for humans without any tools.
> >
> > Also, I suggest that for the final version, you report the human performance results with more subjects, especially out of the authors' circle.

---

> > > ### Author Response · Authors · 2025-11-25
> > >
> > > Thank reviewer for the positive feedback and for raising your score. We truly appreciate your careful examination of our rebuttal and the newly added experiments.
> > >
> > > **Regarding human performance on the object grounding task:** In the response to Q4, we mentioned that for the grounding task in our human evaluation, we used LabelImg [1], a lightweight annotation tool. Rest assured, we will clearly include this information in the final version:
> > >
> > > As humans do not naturally produce pixel-level bounding box coordinates when viewing diagrams, our human evaluation used tool-assisted annotation. Specifically:
> > > - Annotators used LabelImg, which enables drawing bounding boxes directly on the diagram.
> > > - The tool automatically outputs precise pixel coordinates, ensuring full consistency with the evaluation protocol used for models.
> > > - Human performance was evaluated using the same IoU-based metric as in the model evaluation.
> > >
> > > Thus, humans performed bounding-box drawing, and the tool returned the numerical coordinates used for evaluation.
> > >
> > > **Regarding your suggestion on expanding human evaluation:** We fully agree that using only the authors’ team for the human study is not sufficient. We will include additional human participants outside the author group to strengthen the reliability and generality of the human performance statistics in the final version.
> > >
> > > Thank you again for your thoughtful follow-up question and constructive suggestions.
> > >
> > > > [1] https://github.com/tzutalin/labelImg

---

### Author Response · Authors · 2025-11-20
**Paper Revision**

## We sincerely thank all reviewers for their thorough, constructive, and high-quality feedback.

Your comments greatly helped us refine the manuscript. We have carefully revised the paper in response to each reviewer’s suggestions, and all newly added content is highlighted in blue in the revised version. Below, we summarize the key changes.


### Key Revisions Made as Suggested by Reviewers:

1. Added two new ablation tables (Table 11 and Table 12), including the numeric-marker ablation that rules out vertex-count shortcuts in shape classification. *→ As suggested by Reviewer Y1aj*

2. Added a new figure (Figure 8) illustrating numeric-marker examples used in the shortcut ablation. *→ As suggested by Reviewer Y1aj*

3. Updated figures for improved clarity, including the radar plot in Figure 1, revised visualizations in Figure 5 and Figure 6 with an updated color palette for better readability. *→ As suggested by Reviewer Y1aj*

4. Analyzed our results through the lens of the valuable work Visual Graph Arena (VGA) and incorporated a corresponding discussion in the Related Work section. *→ As suggested by Reviewer Y1aj*

5. Added new evaluations of Vision-R1 and MINT-CoT on MATHEMETRIC, and included our model’s results on MathVision in Table 1 and Table 2. *→ As suggested by Reviewer DjMf*

6. Added an in-depth discussion of recent advances in perception–reasoning integration (MINT-CoT and Vision-R1). These approaches align with our finding that perception supports reasoning, yet they operate at different scopes. *→ As suggested by Reviewer DjMf*

7. Added detailed explanation of the design and new experiments with explicit modality-preference instructions (Table 5), e.g., visual-priority vs. text-priority, to strengthen the causal analysis of the “blind faith in text” phenomenon. *→ As suggested by Reviewers DjMf and 4XJK*


8. Added detailed analysis of how enhanced perception improves faithful reasoning chain-of-thoughts (lines 1231-1238), using qualitative model responses to demonstrate how GEOMETRIC corrects key perceptual errors and stabilizes multi-step reasoning.  *→ As suggested by Reviewer 4XJK*

Best regards,

Authors of paper 2014

---

### Meta-Review · Area_Chair_6CMt · 2026-01-08

**Summary:**

This paper focuses on the capability of multimodal large language models (MLLMs) to perceive mathematical diagrams. It proposes a benchmark consisting of several tasks related to basic perception. The results show that failures in low-level perception can lead to poor reasoning performance. The paper also presents a training dataset based on graphs of geometric primitives to improve model performance.

Reviewers raised concerns about potential shortcuts in the experimental settings, the lack of additional analyses, human evaluation, and the inclusion of newer baselines and benchmarks. Most of these concerns were properly addressed by the authors through additional experimental results. Based on this, I believe this work is worthy of acceptance.

**Reviewer Concerns:**

- (Fully addressed) Reviewers had concerns about the settings of some experiments that might cause shortcuts. The authors properly addressed these concerns by providing new experiments following the reviewers’ suggestions.
- (Fully addressed) Reviewers had concerns about the human evaluation baseline. The authors provided human evaluation results during the rebuttal.
- (Not addressed) Reviewers had concerns that the chosen tasks are overly simplified and suggested more complex tasks, such as topological reasoning, implicit constraints, and multi-object composition.
- (Partially addressed) Reviewers had concerns about not including recent models as baselines and not including MathVision benchmark. The authors provided some additional results during the rebuttal, but not complete results across all configurations, baselines, and datasets.
- (Fully addressed) Additional analysis of the “blind faith in text” phenomenon.
- (Fully addressed) Minor questions regarding the clarity of the paper were properly addressed by the authors.

**Reviewer Scores:**

Based on my evaluation:
- Reviewer Y1aj: 6 -> 8
- Reviewer DjMf: keeps 6
- Reviewer 5MW7: 4 -> 6
- Reviewer 4XJK: 6 -> 8

---

### Decision · Program_Chairs · 2026-01-26

Accept (Poster)